# ToProVAR: Efficient Visual Autoregressive Modeling via Tri-Dimensional Entropy-Aware Semantic Analysis and Sparsity Optimization

**Jiayu Chen**[1]    **Ruoyu Lin**[2]    **Zihao Zheng**[1]    **Jingxin Li**[2]
**Maoliang Li**[1]    **Guojie Luo**[1,3]    **Xiang Chen**[1]*
[1] School of Computer Science, Peking University
[2] School of Electronics Engineering and Computer Science, Peking University
[3] National Key Laboratory for Multimedia Information Processing, Peking University

## Abstract

Visual Autoregressive (VAR) models enhance generation quality but face a critical efficiency bottleneck in later stages. In this paper, we present a novel optimization framework for VAR models that fundamentally differs from prior approaches such as FastVAR and SkipVAR. Instead of relying on heuristic skipping strategies, our method leverages attention entropy to characterize the semantic projections across different dimensions of the model architecture. This enables precise identification of parameter dynamics under varying token granularity levels, semantic scopes, and generation scales. Building on this analysis, we further uncover sparsity patterns along three critical dimensions—token, layer, and scale—and propose a set of fine-grained optimization strategies tailored to these patterns. Extensive evaluation demonstrates that our approach achieves aggressive acceleration of the generation process while significantly preserving semantic fidelity and fine details, outperforming traditional methods in both efficiency and quality. Experiments on Infinity-2B and Infinity-8B models demonstrate that ToProVAR achieves up to 3.4× acceleration with minimal quality loss, effectively mitigating the issues found in prior work. Our code will be made publicly available.

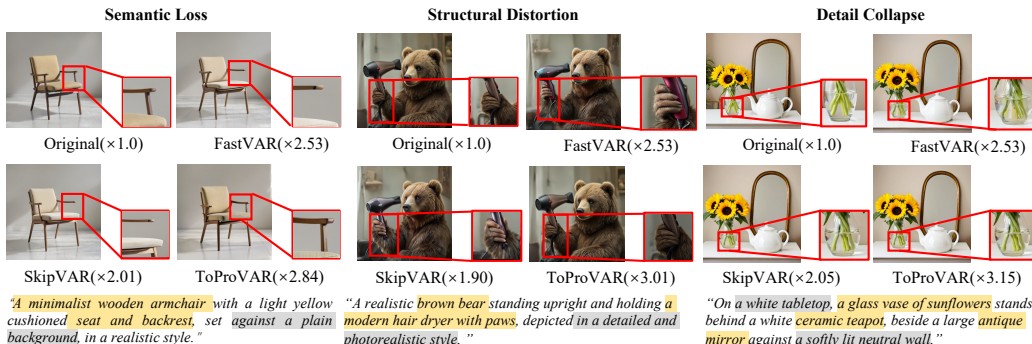

Figure 1: A comparison between our method and state-of-the-art compression methods. The SOTA methods often suffer from issues such as semantic loss, structure distortion, and detail collapse.

## 1 Introduction

Traditional autoregressive (AR) models generate images via raster-scan next-token prediction (Li et al., 2024b; Liu et al., 2024; ai et al., 2025; Xie et al., 2024), which has long produced inferior results compared to diffusion models. Visual AutoRegressive modeling (VAR) (Tian et al., 2024;

---

*Corresponding author.

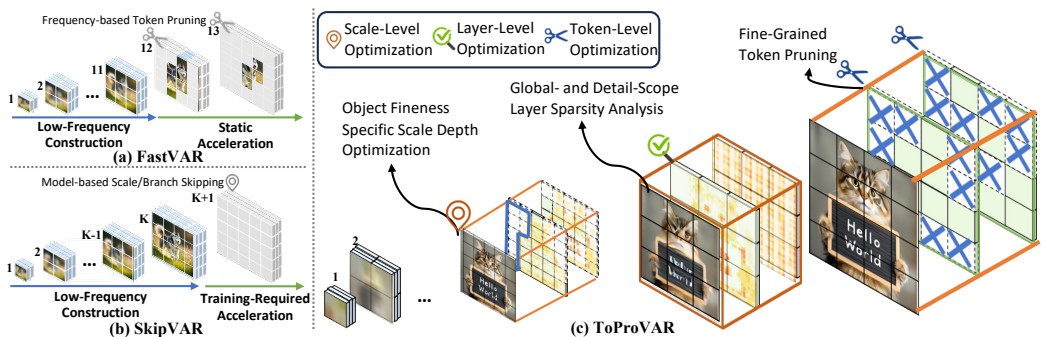

Figure 2: Different Optimization Dimensions – FastVAR vs. SkipVAR vs. ToProVAR

Han et al., 2024; Tang et al., 2024) reformulates generation as coarse-to-fine next-resolution prediction, enabling GPT-style AR models to, for the first time, surpass diffusion models in image quality. Despite these advancements, VAR-based methods still face a core problem: the number of tokens grows exponentially with image resolution and generation scales, resulting in inefficient computation in later stages.

To improve the computational efficiency of VAR models, existing researchs, such as FastVAR (Guo et al., 2025) and SkipVAR (Li et al., 2025a), have explored various token reduction strategies. As shown in Fig. 2(a)(b), FastVAR retains a fixed ratio of high-frequency tokens in the token dimension, while SkipVAR skips certain scales or replaces unconditional branches in the scale dimension based on a trained decision model. Although these methods have demonstrated significant value in accelerating generation, they mainly rely on single-dimensional sparsity analysis of intermediate image data, which introduces several limitations as illustrated in Fig. 1: (1)Semantic loss: specific tokens corresponding to key objects are pruned when their frequency in the image is too low; (2) Structural distortion: a single sparsity metric like frequency in FastVAR cannot capture the complex relative relationships among objects, leading to noticeable deformations in complex regions; (3) Detail collapse: fine-grained objects typically require deeper generation scales for adequate support, while methods like SkipVAR that skip entire scales result in severe loss of details.

Based on the above issues, we identify several critical challenges in optimizing VAR models: (1) Fine-grained sparsity analysis: Unlike prior work, we need to design a highly fine-grained approach to sparsity analysis that effectively prevents information loss caused by misalignment between sparsity metrics and image semantics. (2) Multi-dimensional representation: It is essential to analyze the model across multiple dimensions, enabling not only the assessment of individual token importance but also the accurate characterization of their relative relationships in other dimensions. (3) Efficiency-preserving optimization: While pursuing fine-grained optimization, the analysis itself must remain efficient; otherwise, excessive overhead in modeling sparsity would compromise the overall benefits of the optimization.

To address the aforementioned challenges, we introduce a novel optimization framework – **ToProVAR** – for VAR modeling computing optimization.

First, unlike prior approaches such as FastVAR or SkipVAR, which evaluate sparsity directly on intermediate image representations, we leverage attention entropy to analyze how semantics are projected within the model structure during generation. This enables precise tracking of dynamics under varying object salience, semantic scope, and fineness, thereby providing principled guidance for joint semantic–sparsity analysis and optimization.

Second, we extend entropy-based analysis beyond tokens to cover three complementary dimensions: token-level, layer-level, and scale-level. This multi-dimensional perspective allows us to uncover semantic distributions and correlations that govern sparsity patterns along token, layer, and scale dimensions. As illustrated in Fig. 2(c), this facilitates a series of fine-grained optimizations: token-level pruning of non-essential semantics, layer-level compression that distinguishes global from detail representation, and scale-level disentanglement and depth adjustment of generation tailored to object fineness necessity.

Finally, we design a coordinated optimization algorithm that integrates these three dimensions, while further improving the efficiency of attention-entropy analysis itself. This ensures not only the effectiveness but also the practicality of our framework for real-world VAR acceleration.

We evaluated ToProVAR on mainstream VAR models, Infinity-2B and Infinity-8B. The experimental results show that, compared to FastVAR and SkipVAR, ToProVAR improves inference speed to nearly 3.5× with almost no loss in image quality. In Fig. 1, we demonstrate the high-quality visual results generated by ToProVAR based on the Infinity-8B model, effectively addressing issues such as semantic loss, structural distortion, and detail collapse.

## 2 PRELIMINARY

**Visual Autoregressive Modeling.** VAR redefines the traditional autoregressive paradigm, shifting the core from "next-token prediction" to "next-scale prediction." For a given image, VAR first obtains a feature map through an encoder, which is then quantized into $K$ multi-scale token maps $\mathcal{R} = \{r_1, r_2, \ldots, r_K\}$; The resolutions of these token maps increase progressively according to a scale schedule $\{(h_1, w_1), (h_2, w_2), \ldots, (h_K, w_K)\}$. Each token map $r_k \in \{1, \ldots, V\}^{h_k \times w_k}$ contains $h_k \times w_k$ discrete tokens, all from a codebook of size $V$.

The joint probability distribution over the multi-scale token maps is factorized autoregressively as:

$$p(r_1, r_2, \ldots, r_K) = \prod_{k=1}^{K} p(r_k | r_1, r_2, \ldots, r_{k-1}). \tag{1}$$

Specifically, at each autoregressive scale $k$, the previously generated token maps $\{r_1, \ldots, r_{k-1}\}$ are processed by $L$ layers of the VAR network to predict the current token map $r_k$. This multi-scale generation process supports parallel decoding of multiple tokens within a single token map, significantly improving efficiency compared to traditional "token-by-token" autoregressive models.

**Attention Entropy for Semantic Projection Analysis.** Attention entropy quantifies how concentrated the attention distribution is for a given query. A low entropy value indicates that a token focuses its attention on only a few targets, suggesting strong semantic selectivity; conversely, high entropy reflects a more uniform distribution over targets, implying weaker semantic focus. Formally, given a query $q_i$ and keys $k_j$, the attention weights are defined as scaled dot-products, and the entropy is computed as:

$$\mathcal{H}(q_i) = -\sum_{j=1}^{N} \alpha_{i,j} \log \alpha_{i,j}, \quad \alpha_{i,j} = \frac{\exp(q_i^\top k_j / \sqrt{d_k})}{\sum_{l=1}^{N} \exp(q_i^\top k_l / \sqrt{d_k})}, \tag{2}$$

where $q_i$ and $k_j$ denote the query and key vectors, and $d_k$ is their dimension. Previous studies (Cheng et al., 2022; Pardyl et al., 2023; Choi et al., 2024) have exploited this property of attention entropy to distinguish between foreground and background regions. In this work, we build upon this intuition and employ attention entropy for a different purpose: performing fine-grained semantic projection evaluation of VAR models.

Unlike frequency-based fixed scopes, however, the range of $j$ can be flexibly set across different model structures. Extending $j$ beyond local tokens to encompass cross-layer and multi-scale representations enables entropy to jointly capture semantic evolution across layers and scales. This allows us not only to analyze variations between adjacent tokens, but also to extend the activation range across layer and scale dimensions. In this way, we can simultaneously preserve fine-grained token-level analysis while examining sparsity from broader perspectives in the generation process.

## 3 TRI-DIMENSIONAL ATTENTION ENTROPY GENERALIZATION AND RELATED SEMANTIC AND SPARSITY ANALYSIS

We leverage attention entropy as a unified measure for semantic projection evaluation, enabling the analysis of data sparsity from three dimensions in VAR models.

**Token-Level Attention Entropy for Semantic Salience Analysis.** Previous approaches, such as frequency-based FastVAR, often overlook semantic information; as a result, tiny and fine details

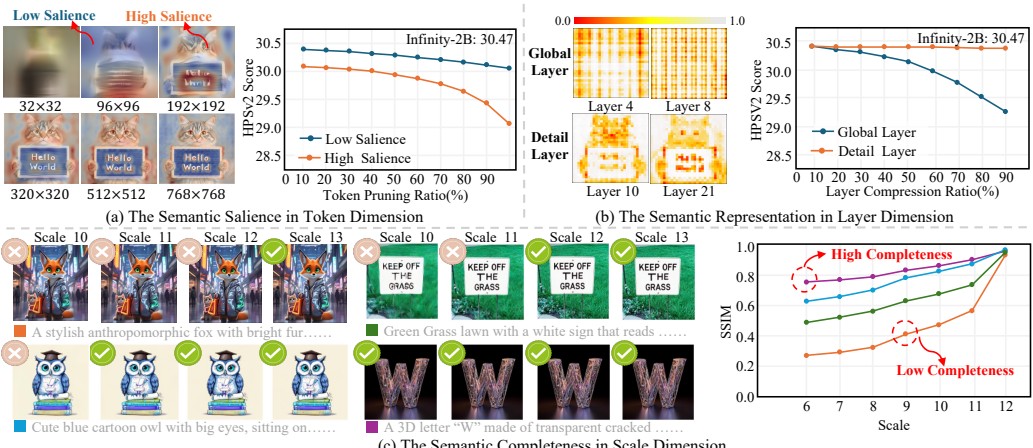

Figure 3: Tri-dimensional attention entropy analysis in VAR models: (a) **Token-Level Semantic Salience**: Pruning low-saliency tokens preserves quality, while pruning high-saliency ones causes severe degradation. (b) **Layer-level Semantic Representation**: Global Layers capture structure and are pruning-sensitive, whereas *detail layers* refine local semantics and can be pruned. (c) **Scale-level Semantic Depth**: complex objects require deeper scales for fine details, while simple ones stabilize earlier, enabling adaptive depth pruning.

are easily over-optimized and consequently lost, as illustrated in Fig. 1. In contrast, by employing attention entropy, our analysis is performed at the model dimension, which enables us to effectively capture generation semantics even for fine-grained content. This substantially improves both controllability and accuracy in generation optimization. Frequency-based approaches typically rely on local averaging within a region. When most of the region contains low-frequency content but only a small portion carries high-frequency details, the averaging process suppresses the latter, leading to their removal during pruning (e.g., the cat's paw in the figure). By leveraging semantic cues, our semantic-based method overcomes this limitation and faithfully preserves both local details and global structures, even under high token pruning ratios.

Therefore, semantic salience refers to the salience distribution across tokens, where only a subset of tokens carries critical semantic information. To further evaluate the impact of semantic salience on quality, we progressively pruned tokens from regions of different salience and measured image quality. The results, shown in the right of Fig. 3(a), demonstrate that on the Infinity-8B model, removing up to 90% of low-saliency tokens leads to only a slight decline in image quality (a drop of <2%). However, pruning just 10% of high-saliency tokens causes a significant loss in generation quality, resulting in noticeable artifacts and semantic gaps.

Based on this exploration of semantic salience, we propose prioritizing the pruning of low-saliency token regions during the later stages of generation. This approach can accelerate the generation process while preserving the quality of the output.

**Layer-level Attention Entropy for Semantic Scope Analysis.** Attention entropy not only enhances granularity at the token level, but its flexible analysis scope also enables semantic patterns to be examined across broader dimensions. By extending the scope of the attention entropy (the range of index $j$ in Eq. 2) to include all tokens within a layer, we can analyze token distributions in the layer dimension and characterize the generation scope of each layer. Specifically, as shown in Fig. 3(b), certain layers exhibit relatively uniform, grid-like attention distributions with prominent principal components, focusing on capturing broad spatial relationships and establishing the overall image structure at a global scope. In contrast, the other layers display more varied, semantic-driven attention patterns with less distinct principal components, concentrating on progressively refining local semantics and fine-grained textures within a localized scope.

This observation motivates distinguishing layers according to their semantic scope, categorizing them as Global Layers and Detail Layers. *Global layers*, with their globally distributed attention, encode strong interconnections across the entire image, whereas *detail layers*, with more local-

Figure 4: Tri-Dimensional Entropy-Aware VAR Sparsity Optimization: (a) **Scale-level**: compute the low-entropy ratio $\rho_s$ across scales and select the pruning start depth via threshold $\tau$. (b) **Layer-level**: for each scale, perform SVD on the entropy map to separate *Global* layers from *Detail* layers. (c) **Token-level**: within prunable layers/scales, increase pruning rate with scale and use entropy-based gating $p_{\text{prune}}$ to remove low-salience tokens, preserving salient regions.

ized attention, concentrate on specific regions, leaving substantial sparsity in the unattended areas. Fig. 3(b) further illustrates the impact of semantic scope on generation quality: compressing *global layers* by more than 50% significantly degrades output quality, whereas even aggressive compression of *detail layers*—up to 90% on the Infinity-8B model—maintains high fidelity. This indicates that selectively pruning *detail layers* can accelerate generation with minimal quality loss.

Implementing this strategy, however, requires a reliable method to distinguish global from detail layers. To this end, we propose to identify global and detail layers by the prominence of principal components in their attention entropy distributions and then prune only the *detail layers*.

**Scale-level Attention Entropy for Semantic Fineness Analysis.** VAR models generate images across multiple autoregressive (AR) scales. Prior approaches often applied coarse-grained scale reduction to reduce computation, but this came at the cost of fine-grained quality. As shown in Fig. 1, images containing fine details require optimization at the semantic granularity level rather than purely at the scale level.

Extending the scope of attention entropy (Eq. 2) beyond local tokens to multiple scales enables us to capture the semantic evolution across scales. Specifically, images with high fineness, such as a complex"cyber fox", exhibit predominantly low-salience distributions, requiring deeper scales to render fine details. In contrast, simpler objects like the letter "W" show predominantly high-salience distributions, stabilizing at shallower scales. As shown in Fig. 3(c), this trend is further corroborated by their SSIM curves.

This scale-level evaluation of attention entropy thus provides a principled way to distinguish the fineness requirements of different objects and adapt the depth accordingly. Based on this observation, we propose dynamically determining the starting scale for pruning according to semantic fineness, enabling adaptive depth allocation for different generation tasks.

## 4 TRI-DIMENSIONAL VAR OPTIMIZATION

Based on our tri-dimensional generalization of attention entropy and the corresponding semantic analysis, we propose a comprehensive framework with optimization techniques for each dimension. As illustrated in Fig. 4, given the multi-scale token maps of a VAR model, $\mathcal{R} = \{r_1, r_2, \ldots, r_k\}$, the framework optimizes the generation process across three dimensions. Specifically, it first estimates the semantic fineness of the scales $\mathcal{R}$ to dynamically determine the scale depth for pruning, $r_i$. Subsequently, within applicable scales, it analyzes each layer's semantic scope to distinguish between Global and Detail layers, pruning only the latter. Finally, fine-grained token-level sparsification is performed within these layers based on attention entropy, with a unified gating function $G$ integrating information from all dimensions to determine the pruning probability for each token.

**Scale-level optimization via Semantic Fineness.** Drawing from our scale-level analysis, as the scale increases, the number of high-salience tokens (orange) gradually decreases, implying fewer tokens need to be processed while the semantic fineness of the generated image improves. To capture

this effect, we quantify the distribution of high-salience tokens across scales by the proportion of low-entropy tokens: a higher proportion indicates finer semantics. As shown in Fig. 4(a), we define the low-entropy ratio at scale $s$ as

$$\rho_s = \frac{\left| i \mid H_i^s < \overline{H}^s \right|}{N_s},$$

(3)

where $H_i^s$ denotes the attention entropy of token $i$ at scale $s$, $\overline{H}^s$ denotes the mean entropy at scale $s$, and $N_s$ denotes the total number of tokens.

Based on this measure, we determine the pruning start scale using a threshold $\tau$. Specifically, the scale depth discrimination function is defined as $D = \min s \mid \rho_s \geq \tau$, where $D$ is the minimum scale index at which semantic stability is achieved. To calibrate $\tau$, we conduct multiple pre-sampling experiments. We observe that as generation converges, $\rho_s$ stabilizes, which provides a reliable criterion for dynamically selecting the pruning depth.

**Layer-level optimization via Semantic Scope.** Based on our layer-level analysis, the next optimization step is to distinguish between *Global Layers* and *Detail Layers* according to their semantic scope. As shown in Fig. 4(b), Global Layers exhibit a pronounced dominant component, while *Detail Layers* do not. To quantify this, we apply singular value decomposition (SVD) to the attention entropy map $X$ of each layer, i.e., $X = U\Sigma V^\top$. In Global Layers, the gap between the largest and the second largest singular values in $\Sigma$ is pronounced, whereas Detail Layers lack such dominance. We thus define the *principal component ratio* $\varrho^{(l,s)} = \sigma_1^{(l,s)}/\sigma_2^{(l,s)}$, where $\sigma_1^{(l,s)}$ and $\sigma_2^{(l,s)}$ denote the two largest singular values of the token representation matrix at layer $l$ and scale $s$.

If $\varrho^{(l,s)} \gg 1$, the layer is classified as Global; if $\varrho^{(l,s)} \approx 1$, it is considered Detail. To provide a continuous score for pruning decisions, we further define the *layer representation score*:

$$\mathcal{R}^{(l,s)} = \exp\left(-\beta(\varrho^{(l,s)} - 1)\right), \quad \beta > 0,$$

(4)

where $\mathcal{R}^{(l,s)} \to 1$ for Detail Layers and $\mathcal{R}^{(l,s)} \to 0$ for Global Layers. This score offers a quantitative criterion for layer-level pruning.

**Token-level optimization via Fine-grained Semantic Salience.** After identifying the prunable scales and layers, the final stage is to perform fine-grained token pruning based on semantic salience. The core idea is to eliminate tokens with low salience (i.e., high attention entropy). As illustrated in Fig. 4(c), to establish a consistent pruning basis across different layers and scales, we first normalize the token's attention entropy:

$$\hat{H}_i^{(l,s)} = \frac{H_i^{(l,s)}}{\sum_{j=1}^{N_{s,l}} H_j^{(l,s)}}, \qquad \text{where} \quad \sum_{i=1}^{N_{s,l}} \hat{H}_i^{(l,s)} = 1.$$

(5)

We then integrate this normalized entropy with the layer score $\mathcal{R}^{(l,s)}$ and a monotonic scale factor $\phi(s) = s/S_{\max}$ to define a unified pruning tendency: $q_i^{(s,l)} = \phi(s) \cdot \mathcal{R}^{(l,s)} \cdot \hat{H}_i^{(s,l)}$. This formulation smoothly incorporates all three dimensions, ensuring that tokens with higher entropy, in layers with broader semantic scope, and at deeper scales are more likely to be pruned.

Finally, the pruning tendency is mapped to a retention probability:

$$P_{\text{keep}}(i \mid s, l) = \begin{cases} 1, & \text{if } s < D, \\ 1 - \text{clip}\left(\alpha_{\min} + (\alpha_{\max} - \alpha_{\min}) q_i^{(s,l)}, \, 0, \, 1\right), & \text{otherwise.} \end{cases}$$

(6)

This three-factor integration provides a coherent pruning policy across tokens, layers, and scales, effectively discarding redundant details while preserving semantically critical structures.

**Computational Optimization for Attention Entropy.** The attention entropy is formally defined in Equation 2. A straightforward implementation would require explicitly materializing the full $N \times N$ attention matrix in order to compute row-wise probability distributions and their entropy. However, such an approach is computationally prohibitive in practice, since modern attention implementations such as FlashAttention never instantiate the dense matrix explicitly due to memory and runtime constraints.

Table 1: **Quantitative comparison** on GenEval and DPG. Note, GenEval follows the official protocol without rewritten prompts. Latency is measured on a single GPU with batch size 1.

| Methods | GenEval | | | | DPG | | | | Latency(s)↓ | Speedup |
|---|---|---|---|---|---|---|---|---|---|---|
| | Two Obj. | Position | Color Attri. | Overall↑ | Entity | Relation | Attribute | Overall↑ | | |
| Infinity-2B | 79.01 | 24.00 | 58.00 | 0.69 | 90.81 | 88.19 | 87.89 | 83.41 | 2.10 | 1.0 × |
| +FastVAR | 78.79 | 27.75 | 59.50 | 0.68 | 88.86 | **91.57** | 87.46 | **83.39** | 0.80 | 2.6 × |
| +SkipVAR | 76.77 | 26.50 | 57.50 | 0.67 | **89.30** | 87.07 | 87.01 | 82.94 | 1.10 | 2.0 × |
| +ToProVAR | **78.80** | **29.50** | **62.00** | **0.69** | 87.39 | 88.92 | **90.01** | 83.07 | **0.61** | **3.4 ×** |
| Infinity-8B | 96.97 | 61.00 | 75.00 | 0.83 | 90.92 | 93.57 | 88.83 | 86.68 | 4.86 | 1.0 × |
| +FastVAR | 94.19 | 57.00 | 75.25 | 0.81 | 90.80 | **92.30** | 90.40 | 86.50 | 2.01 | 2.4 × |
| +SkipVAR | 94.94 | 57.50 | **76.50** | 0.82 | 89.71 | 90.52 | 90.02 | 86.44 | 2.11 | 2.3 × |
| +ToProVAR | **94.95** | **61.00** | 76.00 | **0.83** | **91.11** | 90.39 | **91.04** | **86.70** | 1.78 | 2.7 × |

Table 2: **Quantitative comparison** on HPSv2.1 and ImageReward, two human preference benchmarks. Latency is measured on a single GPU with batch size 1.

| Methods | HPSv2.1 | | | | | ImageReward↑ | Latency(s)↓ | Speedup |
|---|---|---|---|---|---|---|---|---|
| | Photo | Concept-Art | Anime | Paintings | Overall↑ | | | |
| Infinity-2B | 29.40 | 30.38 | 31.72 | 30.39 | 30.47 | 0.94 | 1.57 | 1.0 × |
| +FastVAR | 28.86 | 29.90 | 31.12 | 29.92 | 29.95 | 0.92 | 0.62 | 2.5 × |
| +SkipVAR | 29.25 | **30.25** | 31.50 | **30.45** | **30.39** | 0.93 | 0.87 | 1.8 × |
| +ToProVAR | **29.26** | 30.15 | **31.44** | 30.23 | 30.27 | **0.93** | **0.58** | **2.7 ×** |
| Infinity-8B | 29.42 | 31.27 | 32.45 | 30.83 | 30.99 | 1.04 | 5.31 | 1.0 × |
| +FastVAR | **29.87** | 30.42 | 31.80 | 29.89 | 30.24 | 1.02 | 1.97 | 2.6 × |
| +SkipVAR | 29.09 | 30.86 | 32.04 | **30.55** | **30.64** | 1.03 | 2.65 | 2.0 × |
| +ToProVAR | 29.19 | **30.89** | **32.09** | 30.24 | 30.58 | **1.04** | **1.75** | **3.0 ×** |

To address this challenge, we extend the original FlashAttention algorithm with an online entropy computation mechanism, which we refer to as *Flash Attention Entropy*. The key idea is to preserve the memory efficiency of FlashAttention while simultaneously maintaining sufficient statistics for entropy computation. Inspired by the online softmax strategy in FlashAttention, we design an incremental update scheme that avoids forming the full attention matrix. More concretely, recall that the entropy involves terms of the form $x \log x$ over normalized attention scores. By leveraging the algebraic identity $kx \log(kx) = kx \log x + (\log k) \cdot xk$ We can decompose the entropy computation into two accumulative statistics: the standard normalization terms (rowmax $m$ and expsum $l$) that are already tracked in FlashAttention, and an additional intermediate statistic that maintains $x \log x$. This decomposition ensures that the entropy can be computed in a streaming fashion with negligible overhead relative to the baseline FlashAttention kernel. The resulting algorithm, termed *Flash Attention Entropy*, thus inherits the linear-time and memory-efficient properties of FlashAttention while enabling exact entropy computation without approximation.

# 5 EXPERIMENTS

## 5.1 EXPERIMENTAL SETUP

**Models and Evaluation.** We conduct experiments on Infinity-2B and Infinity-8B(Han et al., 2024). We compare ToProVAR with representative approaches such as FastVAR(Guo et al., 2025) and SkipVAR(Li et al., 2025a), while keeping all hyperparameters consistent for fairness. For evaluation, we adopt widely used benchmarks including Geneval(Ghosh et al., 2024), DPG-Bench(Hu et al., 2024), HPSv2(Wu et al., 2023), ImageReward (Xu et al., 2023), and MJHQ30K(Li et al., 2024a). We report Geneval Overall, DPG Overall, HPSv2 score, FID, and CLIP score as quality metrics. Efficiency is assessed in terms of runtime, throughput, and speedup ratio, with latency measured on a single NVIDIA L40 GPU (40GB).

Table 3: **Quantitative comparisons** of FID and CLIP score with different generation categories on the MJHQ30K dataset. Latency is measured on a single GPU with batch size 1.

| Method | Landscape | | | People | | | Food | | |
|---|---|---|---|---|---|---|---|---|---|
| | Latency | FID↓ | CLIP↑ | Latency | FID↓ | CLIP↑ | Latency | FID↓ | CLIP↑ |
| Infinity-2B | 1.67 | 44.1 | 0.267 | 1.71 | 58.91 | 0.281 | 1.69 | 84.2 | 0.270 |
| +FastVAR | 0.60 | 45.1 | 0.264 | 0.61 | 71.8 | 0.274 | 0.61 | 84.7 | 0.273 |
| +SkipVAR | 1.01 | 58.1 | 0.260 | 1.06 | 73.7 | 0.253 | 0.88 | 102.3 | 0.256 |
| +ToProVAR | **0.50** | **44.5** | **0.264** | **0.48** | **58.84** | **0.283** | **0.46** | **84.3** | **0.274** |

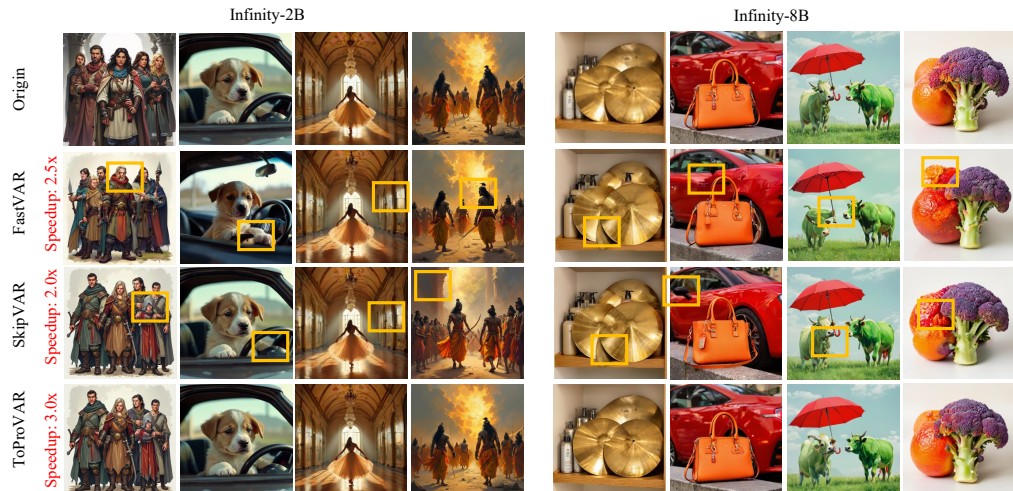

Figure 5: **Qualitative comparison** of various methods on complex prompts. Our method effectively prevents semantic loss, structure distortion, and detail collapse while maintaining high visual fidelity.

## 5.2 MAIN RESULT

**Quantitative Comparison on GenEval and DPG.** We evaluate ToProVAR on GenEval and DPG to assess the quality–efficiency trade-off (Table 1). On Infinity-2B, ToProVAR achieves a 3.4 × speedup while maintaining the same GenEval score as the baseline, and even improves fine-grained metrics such as *Position* (+5.5) and *Color Attribute* (+4.0). On Infinity-8B, it delivers a 2.7 × speedup with no quality degradation, reaching the highest overall scores on both benchmarks. These results show that ToProVAR improves efficiency without sacrificing quality.

**Quantitative Comparison on HPSv2 and ImageReward.** We conduct evaluations on the HPSv2.1 and ImageReward benchmarks to comprehensively evaluate the human preference performance. As shown in Table 2, ToProVAR achieves significant acceleration while maintaining high-quality generation. On the Infinity-2B model, ToProVAR reduces inference latency by 62.4% with a negligible drop in the overall HPSv2 score (<1%). The advantages of ToProVAR are even more pronounced on the larger Infinity-8B model, where it reduces latency by 67% while preserving the same ImageReward score and a highly competitive HPSv2 score. These results powerfully demonstrate that ToProVAR strikes an exceptional balance between efficiency and quality, effectively upholding subjective image quality and human preference alignment while improving inference speed.

**Quantitative Comparison on MJHQ30K.** In Table 3, we validate the perceptual quality on the MJHQ30K benchmark. It can be seen that our ToProVAR achieves a reasonable performance while maintaining a high speedup ratio. For instance, on the challenging People category, our ToProVAR even achieves a FID reduction with 3.5× acceleration, decreasing FID from 58.91 to 58.84. In other categories, our method also maintains good performance with significant acceleration.

**Qualitative Visualizations of Different Methods.** Figure 5 presents qualitative results of different methods on Infinity-2B/8B. Compared to FastVAR and SkipVAR, ToProVAR effectively mitigates

Table 4: Ablation study of the Three-Dimensional Progressive Manipulation Framework on Infinity-2B, where "+", "++", and "+++" denote progressive component addition over the previous stage.

| Method | Latency(s)↓ | Speed ↑ | GenEval↑ |
|---|---|---|---|
| Infinity-2B | 2.10 | - | 0.690 |
| + Scale Depth Loc. | 0.47 | 4.5 × | 0.477 |
| ++ Layer Repr. Ident. | 0.57 | 3.7 × | 0.679 |
| +++ Fine-grained Token Prun. | 0.61 | 3.4 × | 0.690 |

Table 5: Ablation Studies of Flash Attention Entropy on Infinity-2B."w/o FAE" denotes naive attention entropy calculation, which creates computational overhead.

| Setup | Speed↑ | Latency(s)↓ |
|---|---|---|
| Infinity-2B | - | 2.10 |
| +FastVAR | 2.6 × | 0.80 |
| +ToProVAR(w/o FAE) | 1.9 × | 1.10 |
| +ToProVAR(w/ FAE) | 3.4 × | 0.61 |

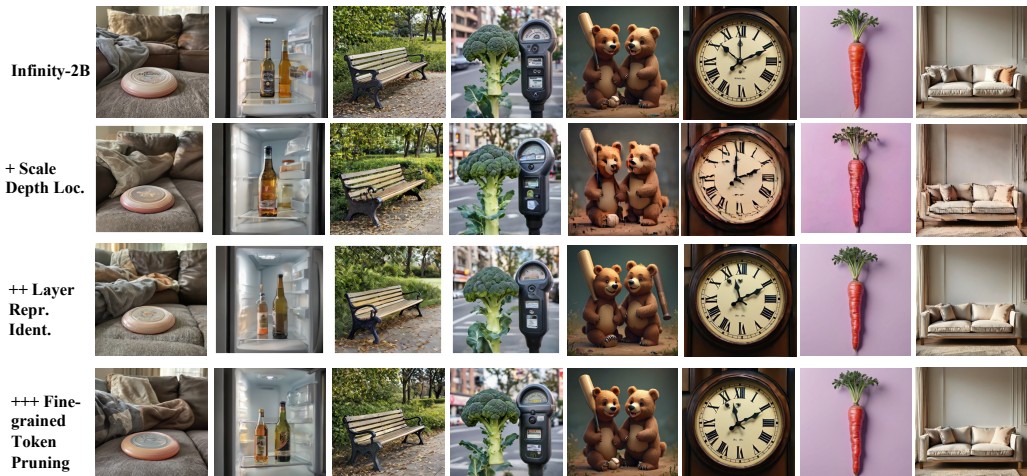

Figure 6: Qualitative Visualizations of the three components ablation(Scale, Layer, Token) on Infinity-2B.The full tri-stage framework better preserves global layout and local details than partial variants that prune along only a subset of dimensions.

semantic loss, structural distortion, and detail collapse. It maintains high visual fidelity while achieving a greater acceleration ratio.

## 5.3 ABLATION STUDY

**Impact of Stages in the Tri-Dimensional Optimization Framework.** To validate the effectiveness of our design, we analyze the contribution of each stage in the framework in Table 4 and Figure 6. A coarse-grained approach using only Scale Depth Localization and Layer Representation Identification is fast but suboptimal, with a GenEval score of 0.679. Incorporating Fine-grained Token Pruning completes the ToProVAR model, restoring the score to 0.690—nearly matching the baseline—while maintaining a low 0.61s latency. This result highlights that our complete tri-stage framework is essential for optimally balancing acceleration and fidelity.

**Ablation Studies on Flash Attention Entropy.** We validate the efficiency of our integrated Flash Attention Entropy (FAE) module in Table 5. Naively attention entropy calculation without FAE creates a significant computational bottleneck, increasing latency to 1.10s. In contrast, our fully integrated module eliminates this overhead and reduces latency to 0.61s. This result confirms that FAE is critical for ToProVAR, as it preserves and enhances the acceleration benefits of Flash Attention.

**Visualization of pruned tokens with different methods.** Fig. 7 visualizes the divergent pruning strategies of FastVAR and ToProVAR. While FastVAR's frequency-based heuristic retains edges, it erodes an image's semantic integrity by removing tokens from smooth yet vital areas like facial contours. This structural damage becomes severe as the pruning ratio increases. In contrast, ToProVAR leverages attention entropy to identify and preserve core semantic content. Even at a 90% prun-

Table 6: Time cost of frequency-, entropy-, and SVD-related operations on Infinity-8B.

| Operation / Time (ms) | All Scales | Rep. Scale s |
|---|---|---|
| *Frequency & Entropy (s = 10)* | | |
| Frequency-based scoring | 1.30 | 0.16 |
| Attention Entropy (naïve) | 125.73 | 12.06 |
| FlashAttention | 11.27 | 1.11 |
| Flash Attention Entropy | 12.97 | 1.28 |
| *Layer-level SVD (s = 6)* | | |
| SVD per layer | – | 0.40 |
| SVD over all layers | – | 15.86 |

Figure 7: Visualization of pruned tokens by Fast-VAR and ToProVAR.

ing ratio, it protects critical features like the eyes. This visu al comparison confirms that attention entropy is a more robust heuristic than frequency for preserving semantic structure during pruning.

**Computational Cost Analysis.** We further quantify the overhead of our tri-dimensional sparsity framework, focusing on Flash Attention Entropy (FAE) and the layer-level SVD analysis. As shown in Table 6, naïve attention entropy incurs a severe bottleneck due to explicit attention-matrix materialization, whereas FAE computes entropy on-the-fly inside the FlashAttention kernel with only a lightweight $x \log x$ reduction. This yields only 0.17 ms overhead over FlashAttention at scale 10, yet reduces entropy cost by $\sim$90%. For layer analysis, SVD is performed once per layer at a single representative scale ($s=6$). Table 6 shows 15.86 ms total over all layers, i.e., $< 1\%$ of the 1780 ms end-to-end latency, indicating minor overhead.

# 6 RELATED WORK

**Autoregressive Visual Generation.** AR models (Li et al., 2024b; Liu et al., 2024; ai et al., 2025; Xie et al., 2024)), originally successful in language, have been extended to image generation through next-token prediction (Van Den Oord et al., 2017; Dai et al., 2025). Recent scaling has narrowed the gap with diffusion models (Xie et al., 2024; Wu et al., 2024b; Wang et al., 2024; Wu et al., 2024a). Yet efficiency remains a bottleneck. To address this, Visual Autoregressive (VAR) modeling(Tian et al., 2024; Han et al., 2024; Tang et al., 2024) introduces a next-scale paradigm, where images are predicted in hierarchical token maps from coarse to fine resolution. This strategy reduces the number of autoregressive steps and improves both speed and quality.

**Efficient Visual Generation.** The acceleration of diffusion models has been extensively studied, with mature methods including distillation (Kim et al., 2025; Zhai et al., 2024; Yin et al., 2025), quantization (Zhao et al., 2024a; Xi et al., 2025; Wu et al., 2025), pruning (Zou et al., 2024; Fang et al., 2025; Heo et al., 2025), and feature caching (Liu et al., 2025b; Zhao et al., 2024b; Ma et al., 2024; Lv et al., 2024; Liu et al., 2025a). Efficient though, they are tailored to diffusion architectures and cannot be directly applied to the hierarchical prediction in VAR.

Efficiency optimization for VAR is still nascent. Early works such as FastVAR (Guo et al., 2025) and SkipVAR (Li et al., 2025a) exploit fixed pruning or frequency-based skipping, while recent methods explore semantic- and structure-aware acceleration, e.g., SparseVAR (Chen et al., 2025a) for token sparsity and CoDe (Chen et al., 2025b) for collaborative decoding. Complementary insights are provided by methods such as HACK (Qin et al., 2025) and ScaleKV (Li et al., 2025b), which focus on KV cache compression.

# 7 CONCLUSION

In this work, we present **ToProVAR**, a novel acceleration framework that addresses tri-dimensional redundancies of Visual Autoregressive models. **ToProVAR** leverages attention entropy to uncover sparsity patterns across tokens, layers, and scales. With fine-grained semantic modeling and pruning strategy, critical contents are preserved even with aggressive acceleration, achieving 3.4× speedup with minimal quality loss, surpassing existing methods. Our study highlights the value of semantic-driven optimization for AR generation and future extensions in video and multimodal modeling.

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
