# ToProVAR: Efficient Visual Autoregressive Modeling via Tri-Dimensional Entropy-Aware Semantic Analysis and Sparsity Optimization

**Jiayu Chen**[1]    **Ruoyu Lin**[2]    **Zihao Zheng**[1]    **Jingxin Li**[2]
**Maoliang Li**[1]    **Guojie Luo**[1,3]    **Xiang Chen**[1*]
[1] School of Computer Science, Peking University
[2] School of Electronics Engineering and Computer Science, Peking University
[3] National Key Laboratory for Multimedia Information Processing, Peking University

## A    Appendix

### A.1    The Use of Large Language Models

Large Language Models (LLMs), specifically ChatGPT, were used as an auxiliary tool in the preparation of this paper. The assistance was limited to polishing English writing and generating small code snippets. No LLMs were used for data generation, experimental results, or research ideation. The authors take full responsibility for all contents of the paper.

### A.2    Code Release and Reproducibility Statement

We provide an anonymized code package that reproduces the main experiments. The package includes: (1) requirements.txt; (2) infer.sh and eval.sh with the exact hyperparameters used for the paper; (3) a README.md with step-by-step reproduction commands; and (4) other code files needed in the paper. For the review process we have provided the anonymized snapshot at the **Anonymous GitHub**. The code is licensed under MIT.

Our experiments were conducted on a machine equipped with **4 L40 GPUs**. The software requirements are managed via a requirements.txt file included in our repository. Additionally, we used the **MJHQ30K(?)** dataset for FID score.

### A.3    Experiment Details

#### A.3.1    Models

Our evaluation is based on the state-of-the-art Visual Autoregressive Models, specifically Infinity-2B and Infinity-8B(**?**). These models have demonstrated exceptional performance across a wide range of image generation tasks. For our experiments, we utilize their pre-trained versions and adopt their default inference configurations, with the prime structures detailed in Table 1.

Table 1: The basic information of models.

| model | scales | layers |
|---|---|---|
| Infinity-2B | 11 | 32 |
| Infinity-8B | 13 | 40 |

#### A.3.2    Baselines Settings

To ensure a fair comparison, we standardized experimental parameters, such as the random seed, across all compared models—Infinity, FastVAR(**?**), SkipVAR(**?**), and our own ToProVAR—to eliminate the influence of confounding variables.

---

*Corresponding author.

We empirically analyzed the hyperparameters of FastVAR and found that its 32 ratio and 40 ratio parameters had a non-monotonic, convex-like effect on generation quality. Consequently, we adopted the optimal, default parameter configuration for our comparisons. For SkipVAR, we directly used its default decision model of (0.84). The specific parameter settings are detailed in the relevant tables.

Table 2: Acceleration Configurations for FastVAR and SkipVAR. NOTE: Notation 's:p' means scale index s is pruned with pruning ratio p,All ratios are applied per-layer at inference time.

| Method | Model | Method Parameter |
|---|---|---|
| FastVAR | Infinity-2B | {10:0.4;11:0.6} |
| | Infinity-8B | {10:0.4;11:0.6;12:1.0;13:1.0} |
| SkipVAR | Infinity-2B | 0.84 |
| | Infinity-8B | 0.84@2B |

### A.3.3 METRICS

In this work, we employ a diverse set of established metrics to comprehensively evaluate our method, aiming to assess both objective image generation quality and adherence to human instructions.

First, we utilize Geneval(**?**) and DPG-bench(**?**) to focus on the objective quantification of generation quality.

**Geneval.** Geneval serves as a crucial tool for measuring foundational generation quality. It decomposes the task into six fine-grained sub-tasks: single-object generation, object co-occurrence, counting, color control, relative positioning, and attribute binding. By using a pre-trained detector to compare generated results with ground-truth annotations, this metric outputs multi-dimensional compliance scores, and its average serves as a comprehensive quality measure.

**DPG-bench.** DPG-bench is a specialized evaluation benchmark for scenarios involving dense prompts. It focuses on the generation quality of multi-object, multi-attribute, and multi-relational descriptions, serving as a key indicator of a model's ability to align with complex semantics and follow instructions.

Second, we leverage HPSv2(**?**), ImageReward(**?**), and SSIM(**?**) to establish a quantitative link between our generated results and human perception, focusing on perceptual similarity and visual preference.

**Human Preference Score v2(HPSv2).** HPSv2 is designed to measure the alignment between generated content and human hierarchical visual perception. It uses a pre-trained visual network to extract both "low-level features (edges, colors)" and "high-level features (object structure, semantics)" from the generated and reference content, then aggregates them to yield a final score. This metric effectively gauges the perceptual plausibility of the generated content from a human perspective.

**ImageReward(IR).** IR is another important metric for aligning text-to-image generation with human preferences. It directly outputs a preference score, quantifying the visual appeal and realism of the generated content.

**Structural Similarity Index Measure (SSIM).** To facilitate the analysis of generation states across images of varying complexity, we introduce SSIM. It provides a measure of image quality that reflects structural and perceptual differences. The formula is defined as:

$$\text{SSIM}(x,y) = \frac{(2\mu_x\mu_y + C_1)(2\sigma_{xy} + C_2)}{(\mu_x^2 + \mu_y^2 + C_1)(\sigma_x^2 + \sigma_y^2 + C_2)}$$

### A.3.4 GENERALIZATION EXPERIMENTS

To assess the generalization of ToProVAR beyond the Infinity series, we further evaluate it on HART **?**, a VAR-style hybrid autoregressive transformer.For HART, we use the official pre-trained checkpoint and default sampling configuration. The resulting quality–efficiency comparison between HART, HART+FastVAR, and HART+ToProVAR is reported in Table 3, showing that ToProVAR maintains comparable GenEval scores to the HART baseline while achieving additional speedups and a better quality–efficiency trade-off than FastVAR.

Table 3: Comparison of ToProVAR and FastVAR on the GenEval benchmark using HART.

| Method | Two Obj. | Position | Color | Attri. | Overall ↑ | Latency (s) ↓ | Speedup ↑ |
|---|---|---|---|---|---|---|---|
| HART | 0.62 | 0.13 | 0.18 | 0.51 | 0.51 | 0.95 | 1.0× |
| +FastVAR | 0.59 | 0.13 | 0.19 | 0.50 | 0.50 | 0.64 | 1.5× |
| +ToProVAR (ours) | 0.61 | 0.13 | 0.18 | 0.51 | 0.51 | 0.56 | 1.7× |

### A.3.5 FURTHER TRI-DIMENSIONAL ABLATION EXPERIMENTS

In the Table **??** of main paper, we provide a three-stage ablation on Infinity-2B. To more clearly isolate the contributions of each dimension in our tri-dimensional framework (Scale → Layer → Token), we further introduce two controlled variants and conduct an extended ablation, summarized in Table 4.

Concretely, all variants share the same Infinity-2B backbone and sampling settings, and differ only in which sparsity dimensions are activated:

- **Fix Scale Exit.** A coarse baseline that skips a fixed set of late scales with a hand-crafted exit scale, without any Scale Depth Localization. This variant achieves the highest nominal speed but suffers from the most severe quality degradation, highlighting the importance of *adaptive* scale selection.

- **Scale Depth Localization (Scale).** A pure scale-skipping variant in which only the Scale Depth Localization module is enabled. It adaptively selects the pruning start scale and skips subsequent scales, improving efficiency over the baseline, but may still introduce noticeable semantic distortions due to the lack of layer- and token-level control.

- **Scale Depth Loc. + Fine-grained Token Pruning (Scale + Token).** A configuration that combines adaptive scale skipping with token-level pruning applied uniformly across *all* layers, without Layer Representation Identification. While this improves over purely scale-level pruning, its quality remains clearly below the baseline, indicating that unconstrained token pruning on Global layers can harm global semantics.

- **Scale Depth Loc. + Layer Representation Identification (Scale + Layer).** A scale–layer variant in which we perform adaptive scale skipping together with layer skipping guided by the layer-scope analysis. Layers identified as Detail are skipped more aggressively, whereas Global layers are largely preserved. This substantially recovers global structure and semantics while maintaining strong acceleration.

- **Full ToProVAR (Scale + Layer + Token).** The complete tri-stage framework, which combines adaptive scale skipping, layer skipping based on Global/Detail classification, and fine-grained token pruning restricted to selected Detail layers. This configuration restores generation quality to be on par with the baseline while retaining substantial speedup.

Overall, these extended ablations demonstrate the *progressive* and *complementary* roles of scale-, layer-, and token-level optimization in balancing acceleration and fidelity: naive scale-only or scale+token pruning can be overly aggressive, whereas the full tri-dimensional design of ToProVAR achieves a much better quality–efficiency trade-off. These configurations correspond to the rows in Table 4 and the visual comparisons in Figure 7, and are used consistently across all ablation experiments.

### A.4 COMPUTATIONAL COST ANALYSIS

In this section, we quantify the computational overhead introduced by our tri-dimensional sparsity framework, focusing on the cost of Flash Attention Entropy (FAE), Singular Value Decomposition (SVD), and the gating operations.

### A.4.1 COST OF ENTROPY COMPUTATIONS.

As shown in Table 5, naïve attention entropy requires full materialization of the attention matrix and is therefore an order of magnitude slower than FlashAttention. In contrast, FAE computes the

Table 4: Extended ablation study of the tri-dimensional progressive framework on Infinity-2B.

| Method | Latency (s) ↓ | Speed ↑ | GenEval ↑ |
|---|---|---|---|
| Infinity-2B | 2.10 | 1.0× | 0.690 |
| Fix Scale Exit | 0.41 | 5.1× | 0.403 |
| Scale Depth Loc. | 0.47 | 4.5× | 0.477 |
| Scale Depth Loc. + Fine-grained Token Prun. | 0.78 | 2.7× | 0.603 |
| Scale Depth Loc. + Layer Repr. Ident. | 0.57 | 3.7× | 0.679 |
| ToProVAR (Scale + Layer + Token) | 0.61 | 3.4× | 0.690 |

Table 5: Time cost of frequency- and entropy-related operations on Infinity-8B. We report the total cost summed over all scales, as well as representative scale 10.

| Operation/Time cost(ms) | All Scales | Scale 10 |
|---|---|---|
| Frequency-based scoring | 1.30 | 0.16 |
| Attention Entropy (naïve) | 125.73 | 12.06 |
| FlashAttention | 11.27 | 1.11 |
| Flash Attention Entropy (FAE, ours) | 12.97 | 1.28 |

entropy on-the-fly inside the FlashAttention kernel (Algorithm 2), without explicitly constructing the attention matrix. The only substantial extra work comes from the row-wise $x \log x$ reduction in the innermost loop, where the $\log$ operation is relatively expensive due to FP64 arithmetic.

As a result, FAE introduces only about 0.17 ms overhead compared to plain FlashAttention at scale 10, while reducing the entropy computation cost by roughly $90\%$ relative to naïve attention entropy. Compared to the overall generation time, this additional overhead is negligible.

### A.4.2 COST OF SVD.

For layer-level analysis, we apply SVD to the attention-entropy maps in order to compute the layer representation score $\mathcal{R}^{(l,s)}$. Concretely, we first average the attention-entropy maps over the head dimension to obtain a single head-aggregated matrix per layer and scale, and then perform SVD on this reduced representation. Importantly, SVD is performed *once per layer at a single scale* (e.g., $s = 6$), rather than at all scales. The measured runtime is reported in Table 6. On Infinity-2B with our ToProVAR framework, the end-to-end latency is about 610 ms. The total SVD time (13.96 ms) thus accounts for less than 3% of the overall inference latency, confirming that the layer representation analysis introduces only a minor overhead in practice.

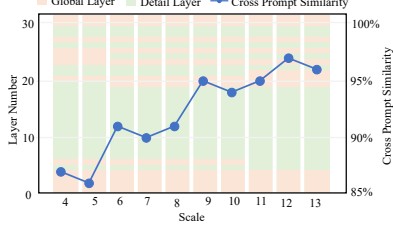

Table 6: Time cost of SVD at a representative scale.

| Operation | Time cost (ms) ↓ |
|---|---|
| SVD per layer at scale $s = 6$ | 0.4650 |
| SVD over all layers at scale $s = 6$ | 13.960 |

Figure 1: Visualization of SVD-based layer analysis.

To justify this design, we further analyze the cross-prompt and cross-scale stability of the SVD-based layer characterization. For each scale, we extract attention-entropy maps on GenEval prompts, perform SVD, and compute the cosine similarity between the resulting singular value spectra across prompts. As illustrated in Fig. 1, the spectra are highly consistent, with cross-prompt similarity exceeding 0.85 at all scales. Using the corresponding layer representation scores, we then classify layers into Global and Detail categories at each scale and visualize their distribution across scales.

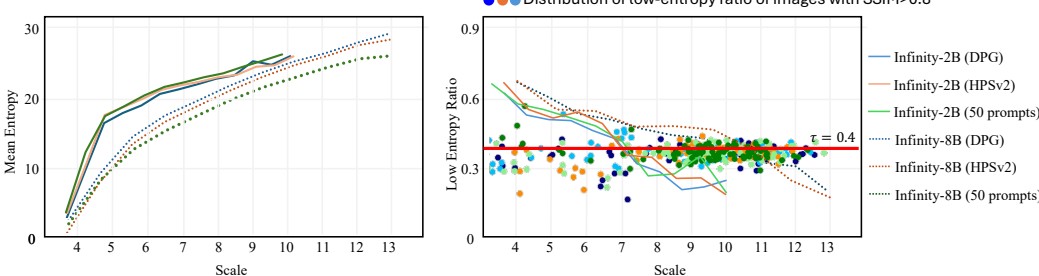

Figure 2: Robustness analysis of the scale-depth threshold $\tau$. Left: mean attention entropy vs. scale. Right: low-entropy ratio $\rho_s$ vs. scale, together with the distribution of $\rho_s$ for images whose SSIM to the full-scale baseline exceeds $0.8$.

The layer labels remain stable in more than $95\%$ of cases, indicating that the semantic roles of VAR layers are largely invariant both across prompts and across scales. These two observations suggest that performing SVD once at a single representative scale yields a layer classification that generalizes well to all scales, making repeated SVD computations unnecessary.

### A.4.3  COST OF TRI-DIMENSIONAL GATING.

The tri-dimensional gating mechanism combines the scale-level depth localization, layer-level representation scores, and token-level attention entropy into a sparse decoding policy. Implementation-wise, this gate consists of simple arithmetic operations and thresholding that are fused into the decoding loop, without additional heavy kernels. Empirically, we observe that its runtime cost is negligible relative to the attention and MLP layers. This explains why, even after accounting for FAE, SVD, and gating overhead, ToProVAR still achieves significant net wall-clock speedups over both the vanilla Infinity models and the frequency-based FastVAR baselines.

### A.5  CALIBRATION AND ROBUSTNESS OF THE SCALE-DEPTH THRESHOLD $\tau$

Building on the scale-level analysis in Eq. (3), we quantify the semantic fineness at scale $s$ by the low-entropy ratio

$$\rho_s = \frac{|\{i \mid H_i^s < \bar{H}^s\}|}{N_s},\tag{1}$$

where $H_i^s$ denotes the attention entropy of token $i$ at scale $s$, $\bar{H}^s$ is the mean entropy at that scale, and $N_s$ is the number of tokens.

To calibrate the pruning start depth, we perform a lightweight pre-sampling procedure on a small set of prompts and compute statistics across scales. For each backbone (e.g., Infinity-2B, Infinity-8B), we randomly sample a calibration subset of prompts from HPSv2 and DPG-Bench, and plot both the *Mean-Entropy–scale* curves and the $\rho_s$–*scale* curves, as shown in Fig. 2. We obtain the following empirical observations:

- **Consistency of mean entropy across datasets.** As shown in the left panel, for a fixed backbone, the mean attention entropy at each scale is highly consistent across HPSv2, DPG-Bench, and a randomly sampled 50-prompt subset: both the absolute values and the scale-wise trends of the curves almost overlap. This indicates that the attention-entropy statistics are largely insensitive to the specific dataset or calibration subset.

- **Stability of the low-entropy ratio across scales.** The right panel shows that the $\rho_s$–scale curves for different datasets and calibration-set sizes (50 prompts vs. the full prompt set) exhibit very similar trajectories. For a given backbone, the scale index at which $\rho_s$ enters a "reasonable" band is highly stable across datasets and prompt subsets.

- **Low-entropy ratio as a quality indicator and choice of $\tau = 0.4$.** The scatter points in the right panel correspond to images whose SSIM with respect to the full-scale baseline is greater than $0.8$. These high-quality generations concentrate around a narrow range of

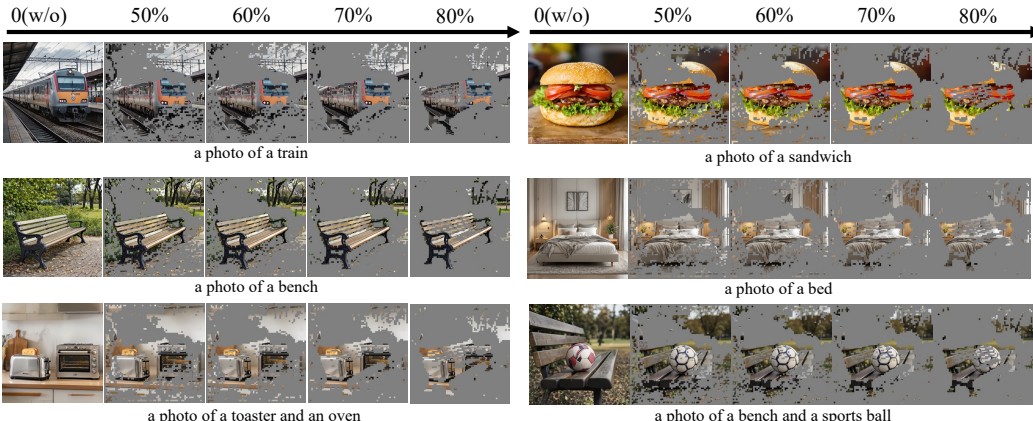

Figure 3: Visualization of token-level pruning in ToProVAR. Gray tokens indicate those pruned by ToProVAR, while colored tokens correspond to preserved, semantically salient regions.

low-entropy ratios, approximately centered at $\rho_s \approx 0.4$. This observation suggests that $\rho_s$ can serve as a proxy for generation quality, and motivates the choice of $\tau = 0.4$ as a quality-aware threshold.

Based on this invariance, we choose a single global threshold $\tau$ per backbone such that $\rho_s \geq \tau$ marks the onset of semantically stable scales suitable for pruning. In practice, $\tau$ is calibrated once on the small calibration subset and then reused across all datasets, resolutions, and prompts, without any dataset-specific hyperparameter search. This explains why the same $\tau$ generalizes well in all our experiments and why our scale-level pruning remains robust under variations in prompts and resolutions.

## A.6 MORE VISUALIZATION RESULTS

This section presents additional visualizations that complement the observations described in the main text.

### A.6.1 PRUNED TOKENS VISUALIZATIONS

In this section, we visualize the token-level pruning decisions made by ToProVAR on image generation tasks with varying levels of complexity. For each example, tokens that are *pruned* by ToProVAR are rendered in gray, while *preserved* tokens retain their original image appearance. As shown in Figure 3, ToProVAR primarily removes tokens in redundant background regions and keeps tokens concentrated around object contours and fine details, leading to more accurate and semantically aligned sparsity patterns.

### A.6.2 LAYER-LEVEL SEMANTIC REPRESENTATION VISUALIZATIONS

In this section, we provide additional visualizations of layer-level semantic representations across different VAR backbones. As shown in Figure 4, we consistently observe two dominant patterns along the layer dimension: some layers exhibit grid-like, globally distributed attention, while others focus on localized, fine-grained regions. This dichotomy underpins our strategy of categorizing layers into Global and Detail layers, and it substantiates the design of Stage II, where semantic analysis and pruning are performed at the layer level.

Crucially, this behavior is not restricted to the Infinity series. When we apply the same layer-scope analysis to HART, we find a very similar organization: early and final layers tend to act as Global Layers with a dominant principal component, whereas middle layers behave as Detail Layers with more localized and diverse semantics. This pattern is stable across prompts and scales when we classify layers at a representative, semantically stable scale.

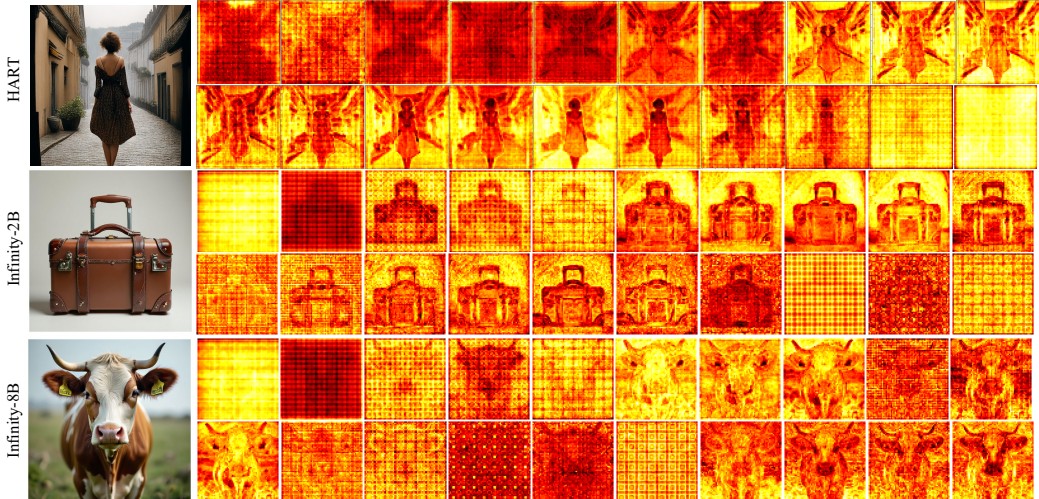

Figure 4: Additional visualizations of layer representations on HART, Infinity-2B, and Infinity-8B.We show example attention maps for the representative Global and Detail layers.

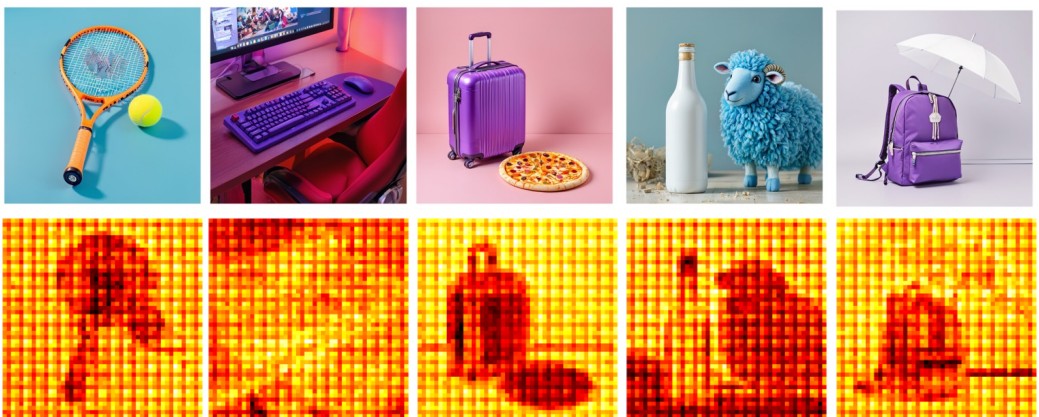

Figure 5: Additional visualizations of failure layer representation case.

Figure 5 further illustrates the rare *ambiguous* cases. In these layers, the attention maps exhibit both weak grid-like global structure and pronounced object-centric activations, so that the layer plays a mixed role of refining global layout and local details. Such hybrid behavior typically arises at transition layers where global composition is being finalized while fine details start to emerge, and is further amplified by averaging over heads, since different heads may specialize in global versus local semantics. Consequently, these layers lie close to the decision boundary of our principal-component–based classifier and can be labeled as Global or Detail depending on small variations across prompts or scales. However, they account for only a very small subset of layers we inspected, and their impact on final accuracy is negligible: our pruning policy is conservative on Detail layers, and the semantics captured by these ambiguous layers are largely redundant with neighboring layers.

### A.6.3 QUALITATIVE COMPARISON OF VARIOUS METHODS

We further present qualitative comparisons of FastVAR, SkipVAR, and ToProVAR on the Infinity-2B and Infinity-8B backbones. As shown in Figure 6, we visualize generated images across diverse prompts and scenes, which allows an intuitive assessment of the quality–efficiency trade-offs achieved by each method. In particular, ToProVAR tends to better preserve global layout and fine-grained details while operating at comparable or higher acceleration levels.

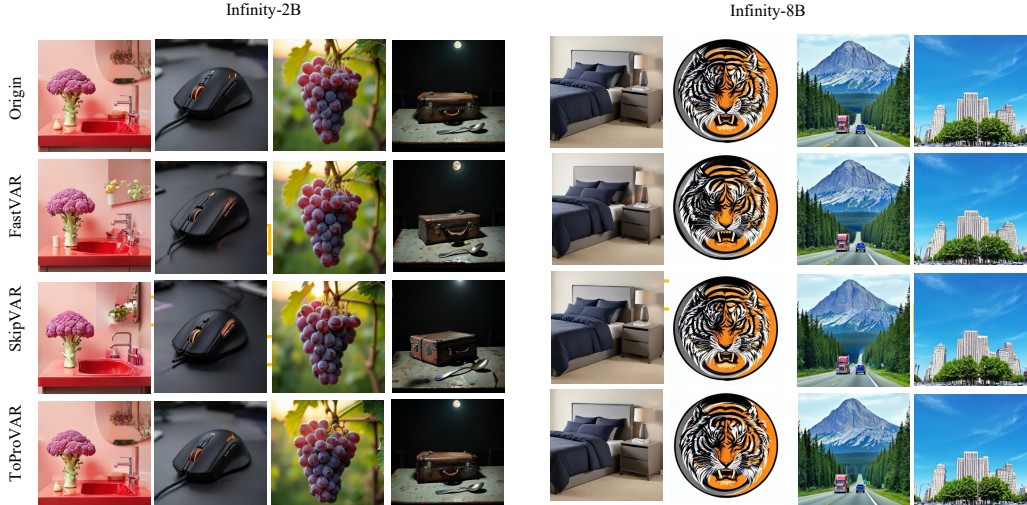

Figure 6: Additional visualization results of FastVAR, SkipVAR, and ToProVAR on Infinity-2B and Infinity-8B. Compared to the baselines, ToProVAR more faithfully preserves global structure and fine details under similar or higher acceleration, yielding visually sharper and more semantically consistent generations.

### A.6.4 VISUALIZATION OF ABLATION STUDY

To further illustrate the impact of the three stages in our tri-dimensional optimization framework(Scale, Layer, and Token), we visualize image generation under different ablation settings. Specifically, in line with the design of Table **??**, we compare three configurations: (i) using only Scale Depth Localization (scale skipping), (ii) adding Layer Representation Identification on top of Scale Depth Localization (Scale + Layer skipping), and (iii) the full ToProVAR framework that additionally incorporates Fine-grained Token Pruning (Scale + Layer + Token).

As shown in Figure 7, relying solely on scale-level decisions yields the strongest acceleration but can introduce structural distortions and missing or blurred objects, especially in semantically complex regions. Introducing layer-level representation identification substantially stabilizes the global layout and object semantics, yet some fine details and textures remain under-optimized. The complete tri-stage configuration (Scale + Layer + Token) recovers both global structure and local details, producing images that are visually close to the baseline while still being substantially faster than the unpruned model. These qualitative results align with the trends observed in the quantitative ablations, indicating that all three stages play complementary roles in balancing sparsity and fidelity.

### A.7 FLASH ATTENTION ENTROPY (FAE)

To efficiently obtain attention entropy without materializing the full attention matrix, we extend the original FlashAttention forward pass to compute entropy on the fly inside the streaming kernel.

As summarized in Algorithm 1, the standard FlashAttention implementation processes $\mathbf{Q}, \mathbf{K}, \mathbf{V}$ in blocks, incrementally updating the partial output $\mathbf{O}_i^{(j)}$ and the normalization statistics $(m_i^{(j)}, \ell_i^{(j)})$ for each query block. At the end of the loop over key/value blocks, it returns the normalized output $\mathbf{O}$ together with the per-row log-sum-exp $L$, which is typically used for numerical stability and backward computation.

Our Flash Attention Entropy (FAE), shown in Algorithm 2, augments this kernel with an additional accumulator $E_i^{(j)}$ that maintains the running sum of $p \log p$ in the same numerically stable streaming fashion used for $\ell_i^{(j)}$. Concretely, we reuse the intermediate unnormalized probabilities $\tilde{\mathbf{P}}_i^{(j)}$ and apply a lightweight `row_reduce_xlogx` operation at each step. After processing all key/value

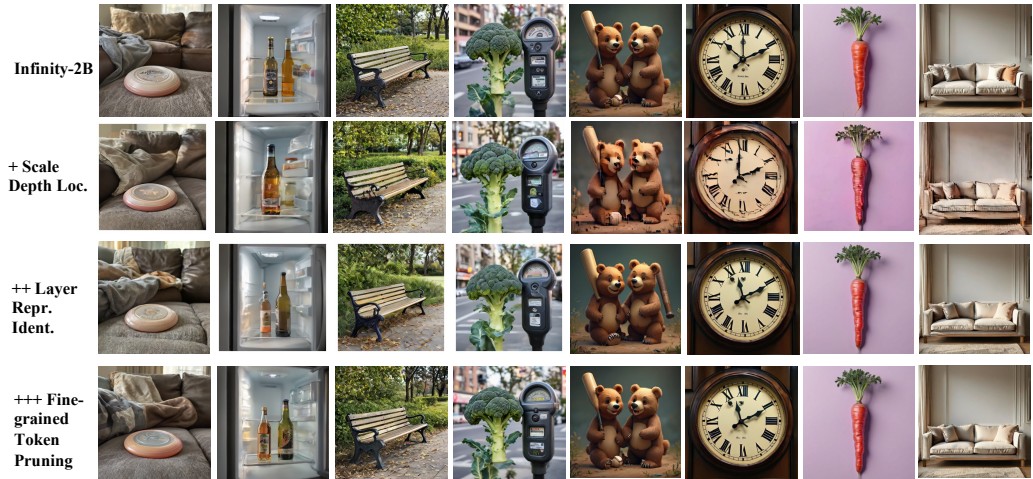

Figure 7: Qualitative ablation of the three components (Scale, Layer, Token) on Infinity-2B.From top to bottom: origin, only Scale Depth Localization, Scale + Layer Representation Identification, and the full ToProVAR with Scale + Layer + Fine-grained Token Pruning. The full tri-stage framework better preserves global layout and local details than partial variants that prune along only a subset of dimensions.

blocks, we obtain the per-query entropy vector $E_i$ by combining $E_i^{(T_c)}$ and $\ell_i^{(T_c)}$, in analogy to how $L_i$ is derived from $m_i^{(T_c)}$ and $\ell_i^{(T_c)}$. This design keeps the memory footprint identical to standard FlashAttention and avoids constructing the full $N \times N$ attention matrix.

Since our sparsity framework only requires the attention output $\mathbf{O}$ and the corresponding entropy $E$ at inference time, we do not use the log-sum-exp $L$ returned by the original kernel. In our implementation, we therefore drop $L$ from the return values and only expose $(\mathbf{O}, E)$, while the core FlashAttention streaming structure remains unchanged.

## A.8 THEORETICAL ANALYSIS OF TOPROVAR

We present a theoretical derivation of the average-case error upper bounds for the tri-dimensional greedy optimization strategy (Scale $\rightarrow$ Layer $\rightarrow$ Token) used in TOPROVAR. All derivations follow the notation and formulae in the main paper, in particular the entropy-based statistics in Equations (2)–(6). Our goal is to show that, under mild assumptions, each stage introduces a bounded error that depends only on a small residual fraction of entropy/importance, so that the overall error remains controlled.

### A.8.1 SCALE-LEVEL ERROR BOUND

We first model generation across scales as an additive refinement process. Let $\mathbf{Z}$ denote the full-resolution representation, and let $\mathbf{Z}_s$ be the representation after completing scale $s$, defined recursively as

$$\mathbf{Z}_s = \mathbf{Z}_{s-1} + \Delta\mathbf{Z}_s, \quad s = 1, \ldots, S_{\max}, \quad \mathbf{Z}_0 = \mathbf{0}, \tag{2}$$

where $\Delta\mathbf{Z}_s$ captures the residual details introduced when moving from scale $s - 1$ to $s$. If we stop refinement at the pruned scale $D$ (determined by the low-entropy ratio $\rho_s$ in Eq. (3)), the pruned representation is

$$\mathbf{Z}_D = \mathbf{Z} - \sum_{s=D+1}^{S_{\max}} \Delta\mathbf{Z}_s, \tag{3}$$

and the scale-level approximation error is

$$E_{\text{scale}} = \|\mathbf{Z} - \mathbf{Z}_D\|_2^2 = \left\| \sum_{s=D+1}^{S_{\max}} \Delta\mathbf{Z}_s \right\|_2^2. \tag{4}$$

Let $\mathcal{S}_{\text{pruned}} = \{\, s \mid D < s \le S_{\max} \,\}$ be the set of pruned scales. Under a standard average-case assumption that the increments $\Delta\mathbf{Z}_s$ are approximately uncorrelated across $s$, we obtain

$$\mathbb{E}[E_{\text{scale}}] \approx \sum_{s \in \mathcal{S}_{\text{pruned}}} \mathbb{E}\big[\|\Delta\mathbf{Z}_s\|_2^2\big]. \tag{5}$$

Empirically, the energy $\|\Delta\mathbf{Z}_s\|_2^2$ is complementary to the low-entropy ratio $\rho_s$ in Eq. (3): scales with a smaller $\rho_s$ retain more high-entropy (less salient) mass and therefore tend to contribute larger residual updates. We model this as

$$\mathbb{E}\big[\|\Delta\mathbf{Z}_s\|_2^2\big] \propto (1 - \rho_s), \tag{6}$$

and denote $F_s = \mathbb{E}\big[\|\Delta\mathbf{Z}_s\|_2^2\big]$ as the expected energy contributed by scale $s$, with total energy $F = \sum_{s=1}^{S_{\max}} F_s = \mathbb{E}\big[\|\mathbf{Z}\|_2^2\big]$.

Following the scale-depth localization in Sec. **??**, we use a low-entropy threshold $\tau$ to decide where to truncate the refinement. Specifically, $D$ is chosen as the smallest scale index such that the low-entropy ratio drops below the threshold,

$$D = \min\{\, s \mid \rho_s \le \tau \,\}, \tag{7}$$

and we prune all subsequent scales $s > D$. In practice, $\tau$ is obtained by a light-weight pre-sampling procedure, and we find that a single value $\tau \approx 0.4$ works robustly across prompts and resolutions.

Empirically, $\rho_s$ is approximately non-increasing for $s \ge D$, so all pruned scales satisfy $\rho_s \le \rho_D \le \tau$ for $s \in \mathcal{S}_{\text{pruned}}$. Therefore

$$\mathbb{E}[E_{\text{scale}}] \approx \sum_{s=D+1}^{S_{\max}} F_s \;\le\; \sum_{s=D+1}^{S_{\max}} (1 - \rho_s)\, F_s \cdot \frac{1}{1 - \rho_D}$$

$$\le (1 - \rho_D) \sum_{s=D+1}^{S_{\max}} \frac{F_s}{1 - \rho_D} \;\le\; (1 - \rho_D)\, F. \tag{7}$$

Since $\rho_D \le \tau$ and $\tau$ is fixed by pre-sampling, Eq. (7) shows that the average scale-level error is controlled by the chosen low-entropy threshold: when the refinement is truncated, only scales whose residual entropy mass is regulated by $\tau$ are dropped.

### A.8.2 LAYER-LEVEL ERROR BOUND

At a fixed scale $s$, the decoder consists of $L$ stacked layers. Let $\mathbf{h}_\ell^{(s)}$ denote the hidden representation after layer $\ell$:

$$\mathbf{h}_0^{(s)} = \mathbf{Z}_s, \quad \mathbf{h}_\ell^{(s)} = f_\ell\big(\mathbf{h}_{\ell-1}^{(s)}\big), \quad \ell = 1, \dots, L. \tag{8}$$

We define the layer-wise residual contribution as

$$\Delta_\ell^{(s)} = \mathbf{h}_\ell^{(s)} - \mathbf{h}_{\ell-1}^{(s)}, \tag{9}$$

so that the full output at scale $s$ can be expressed as

$$\mathbf{h}_L^{(s)} = \mathbf{h}_0^{(s)} + \sum_{\ell=1}^{L} \Delta_\ell^{(s)}. \tag{10}$$

Let $\mathcal{L}_{\text{pruned}}$ be the set of pruned layers at scale $s$, and $\mathcal{L}_{\text{keep}}$ the retained ones. The pruned output (after removing layers in $\mathcal{L}_{\text{pruned}}$) is

$$\mathbf{h}_L^{(s, \text{pruned})} = \mathbf{h}_0^{(s)} + \sum_{\ell \in \mathcal{L}_{\text{keep}}} \Delta_\ell^{(s)}, \tag{11}$$

and the layer-level error is

$$E_{\text{layer}}^{(s)} = \big\|\mathbf{h}_L^{(s)} - \mathbf{h}_L^{(s, \text{pruned})}\big\|_2^2 = \Big\| \sum_{\ell \in \mathcal{L}_{\text{pruned}}} \Delta_\ell^{(s)} \Big\|_2^2. \tag{12}$$

Assuming that the residuals $\{\Delta_\ell^{(s)}\}$ are approximately orthogonal across layers in expectation, we have

$$\mathbb{E}[E_{\text{layer}}^{(s)}] \approx \sum_{\ell \in \mathcal{L}_{\text{pruned}}} \mathbb{E}\big[\|\Delta_\ell^{(s)}\|_2^2\big]. \tag{13}$$

Based on our layer-level analysis (Sec. **??**), each layer at scale $s$ is assigned a representation score $\mathcal{R}^{(\ell,s)}$ computed from the principal component ratio of its attention-entropy map (Eq. (4)): Global Layers have $\mathcal{R}^{(\ell,s)} \to 0$, while Detail Layers have $\mathcal{R}^{(\ell,s)} \to 1$. Let

$$G_s = \mathbb{E}\big[\|\mathbf{h}_L^{(s)}\|_2^2\big], \quad Z_s = \sum_{\ell=1}^{L} \mathcal{R}^{(\ell,s)}, \tag{14}$$

and model the expected contribution of each layer as a normalized fraction of $G_s$,

$$\mathbb{E}\big[\|\Delta_\ell^{(s)}\|_2^2\big] \approx \frac{\mathcal{R}^{(\ell,s)}}{Z_s} G_s. \tag{15}$$

By design, our greedy strategy prunes only Detail Layers (with large $\mathcal{R}^{(\ell,s)}$) and keeps Global Layers (with $\mathcal{R}^{(\ell,s)} \approx 0$). Substituting the above model into the error expression gives

$$\mathbb{E}[E_{\text{layer}}^{(s)}] \approx \sum_{\ell \in \mathcal{L}_{\text{pruned}}} \frac{\mathcal{R}^{(\ell,s)}}{Z_s} G_s$$

$$= \frac{\sum_{\ell \in \mathcal{L}_{\text{pruned}}} \mathcal{R}^{(\ell,s)}}{Z_s} G_s \equiv \gamma_s G_s, \tag{8}$$

where

$$\gamma_s = \frac{\sum_{\ell \in \mathcal{L}_{\text{pruned}}} \mathcal{R}^{(\ell,s)}}{\sum_{\ell=1}^{L} \mathcal{R}^{(\ell,s)}} \in [0,1] \tag{16}$$

measures the fraction of the layer representation score that is discarded at scale $s$. Since Global Layers contribute negligibly to $\mathcal{R}^{(\ell,s)}$ and are always retained, $\gamma_s$ remains small in practice, and the average layer-level error at scale $s$ is linearly controlled by $\gamma_s$ through Eq. (8).

### A.8.3 TOKEN-LEVEL ERROR BOUND

After determining scales and layers, token pruning operates within the remaining Detail layers using entropy-based gating. Let $\mathbf{t}_i$ be the token vector at index $i$, and let $w_i = \widehat{H}_i^{(l,s)}$ be its normalized entropy-based importance (Eq. (5)), satisfying $\sum_i w_i = 1$. Let $\mathcal{T}_{\text{keep}}$ and $\mathcal{T}_{\text{pruned}}$ denote the sets of kept and pruned tokens, respectively.

The full token energy is

$$H = \sum_{i=1}^{N} \|\mathbf{t}_i\|_2^2, \tag{17}$$

and the token-level error introduced by pruning is

$$E_{\text{token}} = \sum_{i \in \mathcal{T}_{\text{pruned}}} \|\mathbf{t}_i\|_2^2. \tag{18}$$

Assuming that token energy is approximately proportional to importance, i.e.,

$$\mathbb{E}\big[\|\mathbf{t}_i\|_2^2\big] \approx w_i H, \tag{19}$$

we obtain the average-case bound

$$\mathbb{E}\big[E_{\text{token}}\big] \approx H \sum_{i \in \mathcal{T}_{\text{pruned}}} w_i. \tag{20}$$

Define the pruned importance mass

$$\gamma = \sum_{i \in \mathcal{T}_{\text{pruned}}} w_i, \tag{21}$$

which measures how much normalized entropy mass is discarded. Then

$$\mathbb{E}[E_{\text{token}}] \leq \gamma\, H. \tag{9}$$

In our design, the gating function $q_i(s, l)$ (Eq. (6)) and the range $[\alpha_{\min}, \alpha_{\max}]$ jointly enforce that high-importance (low-entropy) tokens are kept and that each region preserves at least an $\alpha_{\min}$ fraction of tokens. This makes $\gamma$ significantly smaller than the raw token sparsity ratio and keeps $E_{\text{token}}$ small.

### A.8.4 TOTAL ERROR AND SAFETY

Finally, we combine the contributions from the three stages. Since the scale-, layer-, and token-level errors affect different structural components and pruning is applied in a nested manner (scale first, then layers, then tokens within selected layers), it is reasonable in the average case to treat these error terms as approximately additive:

$$\mathbb{E}[E_{\text{total}}] \ \leq \ \mathbb{E}[E_{\text{scale}}] + \sum_s \mathbb{E}[E_{\text{layer}}^{(s)}] + \mathbb{E}[E_{\text{token}}]. \tag{22}$$

Using the bounds from Eqs. (7)–(9), we obtain the global bound

$$\mathbb{E}[E_{\text{total}}] \ \leq \ (1 - \rho_D)\, F + \sum_s \gamma_s G_s + \gamma H. \tag{10}$$

Here $\rho_D$ is the low-entropy ratio at the truncation scale $D$, $\gamma_s$ is the discarded layer-level representation fraction at scale $s$, and $\gamma$ is the discarded token-level importance mass.

In our safe operating regime, the threshold $\tau$ and the gating parameters $[\alpha_{\min}, \alpha_{\max}]$ are selected such that the empirical residual fractions $\gamma_s$ and $\gamma$ remain small (see Sec. **??** and Appendix X for empirical ranges). The nested structure further prevents cross-stage amplification: token pruning is only applied after conservative scale- and layer-level decisions, and all three stages include fallback conditions (e.g., no scale pruning if the empirical $\rho_s$ profile does not cross the threshold $\tau$, no layer pruning when the Global/Detail classification is ambiguous, and a minimum per-region token keep ratio).

In summary, while the tri-stage strategy is greedy and heuristic, the entropy-based quantities $(\rho_s, \mathcal{R}^{(\ell,s)}, \widehat{H}_i^{(l,s)})$ and the associated thresholds $(\tau, \alpha_{\min}, \alpha_{\max})$ induce explicit average-case error upper bounds in each dimension, clarifying why errors do not compound in the operating regime used in our experiments.

### A.9 LIMITATIONS AND FUTURE WORK

Despite the promising acceleration and quality preservation results, ToProVAR has several limitations that suggest important directions for future work.

**Limitations**

1. **Architecture Dependency.** The framework fundamentally relies on the **attention mechanism** and the derived attention entropy for semantic analysis, limiting its direct applicability to non-Transformer-based generative models.

2. **Parameter Sensitivity.** Optimal performance requires manual tuning of the proportion of low-entropy tokens, which hinders truly adaptive and zero-configuration deployment.

**Future Work**

1. **Online Adaptive Control.** We will explore RL to learn to **dynamically predict** optimal pruning strategy.

2. **Efficient video generation and editing.** We plan to extend the framework to V-VAR models, achieving efficient 4D semantic projection by incorporating temporal saliency. Furthermore, the fine-grained semantic map might be leveraged for **high-efficiency local image/video editing**.

---

**Algorithm 1** original FlashAttention forward pass

---

**Require:** Matrices $\mathbf{Q}, \mathbf{K}, \mathbf{V} \in \mathbb{R}^{N \times d}$ in HBM, block sizes $B_c$, $B_r$.

1: Divide $\mathbf{Q}$ into $T_r = \left\lceil \frac{N}{B_r} \right\rceil$ blocks $\mathbf{Q}_1, \ldots, \mathbf{Q}_{T_r}$ of size $B_r \times d$ each, and divide $\mathbf{K}, \mathbf{V}$ in to $T_c = \left\lceil \frac{N}{B_c} \right\rceil$ blocks $\mathbf{K}_1, \ldots, \mathbf{K}_{T_c}$ and $\mathbf{V}_1, \ldots, \mathbf{V}_{T_c}$, of size $B_c \times d$ each.

2: Divide the output $\mathbf{O} \in \mathbb{R}^{N \times d}$ into $T_r$ blocks $\mathbf{O}_i, \ldots, \mathbf{O}_{T_r}$ of size $B_r \times d$ each, and divide the logsumexp $L$ into $T_r$ blocks $L_i, \ldots, L_{T_r}$ of size $B_r$ each.

3: **for** $1 \leq i \leq T_r$ **do**

4:   Load $\mathbf{Q}_i$ from HBM to on-chip SRAM.

5:   On chip, initialize $\mathbf{O}_i^{(0)} = (0)_{B_r \times d} \in \mathbb{R}^{B_r \times d}, \ell_i^{(0)} = (0)_{B_r} \in \mathbb{R}^{B_r}, m_i^{(0)} = (-\infty)_{B_r} \in \mathbb{R}^{B_r}$.

6:   **for** $1 \leq j \leq T_c$ **do**

7:     Load $\mathbf{K}_j, \mathbf{V}_j$ from HBM to on-chip SRAM.

8:     On chip, compute $\mathbf{S}_i^{(j)} = \mathbf{Q}_i \mathbf{K}_j^T \in \mathbb{R}^{B_r \times B_c}$.

9:     On chip, compute $m_i^{(j)} = \max(m_i^{(j-1)}, \mathrm{rowmax}(\mathbf{S}_i^{(j)})) \in \mathbb{R}^{B_r}$, $\tilde{\mathbf{P}}_i^{(j)} = \exp(\mathbf{S}_i^{(j)} - m_i^{(j)}) \in \mathbb{R}^{B_r \times B_c}$ (pointwise), $\ell_i^{(j)} = e^{m_i^{j-1} - m_i^{(j)}} \ell_i^{(j-1)} + \mathrm{rowsum}(\tilde{\mathbf{P}}_i^{(j)}) \in \mathbb{R}^{B_r}$.

10:     On chip, compute $\mathbf{O}_i^{(j)} = \mathrm{diag}(e^{m_i^{(j-1)} - m_i^{(j)}})^{-1} \mathbf{O}_i^{(j-1)} + \tilde{\mathbf{P}}_i^{(j)} \mathbf{V}_j$.

11:   **end for**

12:   On chip, compute $\mathbf{O}_i = \mathrm{diag}(\ell_i^{(T_c)})^{-1} \mathbf{O}_i^{(T_c)}$.

13:   On chip, compute $L_i = m_i^{(T_c)} + \log(\ell_i^{(T_c)})$.

14:   Write $\mathbf{O}_i$ to HBM as the $i$-th block of $\mathbf{O}$.

15:   Write $L_i$ to HBM as the $i$-th block of $L$.

16: **end for**

17: Return the output $\mathbf{O}$ and the logsumexp $L$.

---

**Algorithm 2** FlashAttention forward pass with entropy

---

**Require:** Matrices $\mathbf{Q}, \mathbf{K}, \mathbf{V} \in \mathbb{R}^{N \times d}$ in HBM, block sizes $B_c$, $B_r$.

1: Divide $\mathbf{Q}$ into $T_r = \left\lceil \frac{N}{B_r} \right\rceil$ blocks $\mathbf{Q}_1, \ldots, \mathbf{Q}_{T_r}$ of size $B_r \times d$ each, and divide $\mathbf{K}, \mathbf{V}$ in to $T_c = \left\lceil \frac{N}{B_c} \right\rceil$ blocks $\mathbf{K}_1, \ldots, \mathbf{K}_{T_c}$ and $\mathbf{V}_1, \ldots, \mathbf{V}_{T_c}$, of size $B_c \times d$ each.

2: Divide the output $\mathbf{O} \in \mathbb{R}^{N \times d}$ into $T_r$ blocks $\mathbf{O}_i, \ldots, \mathbf{O}_{T_r}$ of size $B_r \times d$ each, and divide the logsumexp $L$ into $T_r$ blocks $L_i, \ldots, L_{T_r}$ of size $B_r$ each.

3: **for** $1 \leq i \leq T_r$ **do**

4:   Load $\mathbf{Q}_i$ from HBM to on-chip SRAM.

5:   On chip, initialize $\mathbf{O}_i^{(0)} = (0)_{B_r \times d} \in \mathbb{R}^{B_r \times d}, \ell_i^{(0)} = (0)_{B_r} \in \mathbb{R}^{B_r}, E_i^{(0)} = (0)_{B_r} \in \mathbb{R}^{B_r}, m_i^{(0)} = (-\infty)_{B_r} \in \mathbb{R}^{B_r}$.

6:   **for** $1 \leq j \leq T_c$ **do**

7:     Load $\mathbf{K}_j, \mathbf{V}_j$ from HBM to on-chip SRAM.

8:     On chip, compute $\mathbf{S}_i^{(j)} = \mathbf{Q}_i \mathbf{K}_j^T \in \mathbb{R}^{B_r \times B_c}$.

9:     On chip, compute $m_i^{(j)} = \max(m_i^{(j-1)}, \mathrm{rowmax}(\mathbf{S}_i^{(j)})) \in \mathbb{R}^{B_r}$, $\tilde{\mathbf{P}}_i^{(j)} = \exp(\mathbf{S}_i^{(j)} - m_i^{(j)}) \in \mathbb{R}^{B_r \times B_c}$ (pointwise), $\ell_i^{(j)} = e^{m_i^{j-1} - m_i^{(j)}} \ell_i^{(j-1)} + \mathrm{rowsum}(\tilde{\mathbf{P}}_i^{(j)}) \in \mathbb{R}^{B_r}$, $E_i^{(j)} = e^{m_i^{j-1} - m_i^{(j)}} E_i^{(j-1)} + \mathrm{rowreducexlogx}(\tilde{\mathbf{P}}_i^{(j)}) \in \mathbb{R}^{B_r}$.

10:     On chip, compute $\mathbf{O}_i^{(j)} = \mathrm{diag}(e^{m_i^{(j-1)} - m_i^{(j)}})^{-1} \mathbf{O}_i^{(j-1)} + \tilde{\mathbf{P}}_i^{(j)} \mathbf{V}_j$.

11:   **end for**

12:   On chip, compute $\mathbf{O}_i = \mathrm{diag}(\ell_i^{(T_c)})^{-1} \mathbf{O}_i^{(T_c)}$.

13:   On chip, compute $E_i = E_i^{(T_c)} (\ell_i^{(T_c)})^{-1} + \log((\ell_i^{(T_c)})^{-1})$.

14:   On chip, compute $L_i = m_i^{(T_c)} + \log(\ell_i^{(T_c)})$.

15:   Write $\mathbf{O}_i$ to HBM as the $i$-th block of $\mathbf{O}$.

16:   Write $E_i$ to HBM as the $i$-th block of $E$.

17: **end for**

18: Return the output $\mathbf{O}$ and the entropy $E$.

---