# OpenReview forum: "ToProVAR: Efficient Visual Autoregressive Modeling via Tri-Dimensional Entropy-Aware Semantic Analysis and Sparsity Optimization"
_ICLR.cc/2026/Conference — ICLR 2026 Poster_

### Official Review · Reviewer_FVpx · 2025-10-28

**Soundness:** 3
**Presentation:** 3
**Contribution:** 3
**Rating:** 8
**Confidence:** 5

**Summary:**

This paper proposes ToProVAR for visual autoregressive models' acceleration. ToProVAR can identify parameter dynamics and redundancy at different granularities, semantic scopes, and generation scales, enabling fine-grained acceleration and pruning. In the experiments, the authors conduct comprehensive quantitative and qualitative evaluations on mainstream VAR models (Infinity-2B and Infinity-8B). Results show that ToProVAR achieves an average 3.5× inference speedup with almost no loss in generation quality, significantly outperforming existing SOTA methods.

**Strengths:**

- To obtain the entropy of the attention map while remaining compatible with FlashAttention, the authors propose FlashAttn-Entropy, which is noteworthy.

- The proposed method achieves faster speed against previous methods like FastVAR.

- The paper is well-written and easy to follow.

**Weaknesses:**

- The proposed method is only evaluated on the Infinity model families. How does it perform on other model families such as HART?

- The proposed method requires searching for the best hyperparameter combinations on different datasets. What is the computational cost of this process?

- In Figure 3, the meaning of the lines with different colors is unclear. It is recommended to add a legend in this figure. Moreover, in Table 1, the bolded performance on the DPG dataset seems to be incorrect (FastVAR’s 83.39 should be better).

**Questions:**

- It seems that the proposed method requires performing online SVD for different samples. What is the time cost of this step?

- What is the computational overhead of the proposed FlashAttn-Entropy?

---

> ### Author Response · Authors · 2025-11-28
> **Response to Reviewer FVpx(1/2)**
>
> We sincerely appreciate your valuable feedback and recognition of our work's contribution. We have carefully addressed each of your questions and detailed our responses below.
>
>
> > **Q1:** The proposed method is only evaluated on the Infinity model families. How does it perform on other model families such as HART?
>
> A1: Thank you for raising this important question about the generality of our method beyond the Infinity model family. To assess the broader applicability of ToProVAR, we further evaluated it on HART [1], a VAR-based model with a markedly different attention design. As shown in the new results on HART (Tables 1 below), ToProVAR maintains stable generation quality on GenEval with only negligible degradation relative to the HART baseline, while significantly reducing inference latency. Moreover, compared with FastVAR, ToProVAR consistently achieves a better quality–efficiency trade-off on HART, indicating that our tri-dimensional entropy-aware sparsity optimization is not tied to Infinity-specific architectures and can be effectively transferred to other VAR-style model families.
>
> **Table 1: Comparison of ToProVAR and FastVAR on the GenEval benchmark using HART.**
>
> | Method        | Two Obj. | Position | Color  Attri. | Overall↑ | Latency (s)↓ | Speedup↑ |
> |---------------|----------|----------|---------------|----------|--------------|----------|
> | HART          | 0.62     | 0.13     | 0.18          | 0.51     | 0.95s        | 1.0×     |
> | +FastVAR      | 0.59     | 0.13     | 0.19          | 0.50     | 0.64s        | 1.5×     |
> | +ToProVAR     | 0.61     | 0.13     | 0.18          | 0.51     | 0.56s        | 1.7×     |
>
> [1] **HART: Efficient Visual Generation with Hybrid Autoregressive Transformer.**

---

> ### Author Response · Authors · 2025-11-28
> **Response to Reviewer FVpx(2/2)**
>
> > **Q2:** The proposed method requires searching for the best hyperparameter combinations on different datasets. What is the computational cost of this process?
>
> > **Q3:** It seems that the proposed method requires performing online SVD for different samples.What is the time cost of this step?
>
> > **Q4:** What is the computational overhead of the proposed FlashAttn-Entropy?
>
> A2: Thank you for your thoughtful questions on the computational cost of hyperparameter tuning, SVD, and FlashAttn-Entropy. We summarize the overhead below.
> 1. **Hyperparameter tuning.**
>    Our method does *not* require searching for different hyperparameter combinations on each dataset. The only scalar hyperparameter is scale-level threshold $\tau$, which is calibrated **once per model** via a lightweight offline pre-sampling procedure. In practice, we use 50 prompts randomly sampled from HPSv2.1/DPG-Bench to estimate the scale-wise low-entropy ratio $\rho_s$ and fix $\tau$. This one-time calibration costs about **0.3 L40 GPU-hours**, which is negligible compared to model training and large-scale evaluation, and the same $\tau$ is reused across all datasets and resolutions (robustness analysis is provided in **Appendix A.5**).
> 2. **Online SVD cost.**
>    In the revised manuscript we clarify that SVD is computed **once per layer at a single representative scale** (e.g., scale $s = 6$), rather than at all scales. **Appendix A.4.2** reports a per-layer SVD time of **0.465 ms**, and a total SVD time of **13.96 ms** at that scale, which accounts for **less than 3%** of the overall inference latency for Infinity-2B + ToProVAR (**$\approx$610 ms**).
>
> 3. **Flash Attention Entropy overhead.**
>    Our Flash Attention Entropy (FAE) computes entropy inside the FlashAttention kernel without explicitly constructing the full attention matrix. The only substantial extra work comes from the row-wise $x \log x$ reduction in the innermost loop.
> As shown in Table 4,  its runtime over all scales on Infinity-8B is **12.96 ms**, compared to **11.27 ms** for plain FlashAttention and **125.73 ms** for a naïve attention-entropy implementation. Thus, FAE adds only a small absolute overhead (1.69 ms) per layer over plain FlashAttention,, while reducing the entropy computation cost by roughly **90%** compared to naïve Attention Entropy.
>
> **Table 3: Time cost of SVD.**
>    | Operation                         | Time cost (ms) ↓ |
>    | --------------------------------- | ---------------- |
>    | SVD per layer at scale 6          | 0.4650           |
>    | SVD (all layers) at scale $s = 6$ | 13.960           |
>
> **Table 4: Time cost of Attention Entropy. (per layer)**
>
>
>    | **Operation/Time cost (ms)**                          | **All Scales** | **Scale 10** |
>    | ----------------------------- | ------------------------------------------- | -------- |
>    | Attention Entropy (naïve)     | 125.73                                      | 12.06    |
>    | FlashAttention                | 11.27                                       | 1.11     |
>    | Flash Attention Entropy (FAE) | 12.97                                       | 1.28     |
>
> We included this detailed cost analysis and the above table in **Appendix A.4 “Computational Cost Analysis”** of the revised manuscript to make the overhead of each component explicit.
>
> > **Q5:** In Figure 3, the meaning of the lines with different colors is unclear. It is recommended to add a legend in this figure. Moreover, in Table 1, the bolded performance on the DPG dataset seems to be incorrect (FastVAR’s 83.39 should be better).
>
> A3: We thank the reviewer for the helpful comments and apologize for the ambiguity in the original figures and tables. In the revised version, we have added a clear legend to Figure 3(c) to clarify the meaning of the different-colored curves (each color now explicitly corresponds to a specific prompt), and we have corrected the boldface formatting in Table 1 so that the highest DPG score is properly highlighted.  We appreciate your careful review, which has helped improve the clarity and accuracy of our manuscript.

---

### Official Review · Reviewer_2xuz · 2025-10-30

**Soundness:** 3
**Presentation:** 3
**Contribution:** 3
**Rating:** 8
**Confidence:** 3

**Summary:**

The paper proposes ToProVAR, a novel acceleration framework for Visual Autoregressive (VAR) models that leverages attention entropy to perform fine-grained sparsity analysis across three dimensions—token, layer, and scale. By dynamically identifying and pruning semantically redundant computations, ToProVAR achieves up to 3.5× inference speedup on Infinity-2B and Infinity-8B models with minimal degradation in generation quality, effectively addressing issues like semantic loss, structural distortion, and detail collapse observed in prior methods such as FastVAR and SkipVAR.

**Strengths:**

1. The idea of employing attention entropy is interesting and the motivation is strong.
2. Solid experiments and promising performance.

**Weaknesses:**

1. It would be beneficial to include results on additional backbones beyond Infinity, such as HART [1] and STAR [2], to demonstrate the generalizability of ToProVAR across different VAR architectures.
2. The paper does not provide a detailed analysis of the computational overhead of SVD. Is there an analysis of the layer representation score? What are the patterns in the emergence of Global Layers and Detail Layers? Figure 10 shows an alternating pattern, is this behavior general?
3. According to Table 4, Layer Representation Identification contributes most significantly to preserving generation quality (GenEval ↑ from 0.477 to 0.679), suggesting its critical role. Thus, more analysis such as qualitative cases would strengthen the paper’s claims.

**Questions:**

See weakness.

---

> ### Author Response · Authors · 2025-11-28
> **Response to Reviewer 2xuz(1/2)**
>
> > **Q1**: It would be beneficial to include results on additional backbones beyond Infinity, such as HART and STAR, to demonstrate the generalizability of ToProVAR across different VAR architectures.
>
> A1: Thank you for raising this important point about the generalizability of our method beyond the Infinity family. To assess the broader applicability of ToProVAR, we additionally evaluate it on **HART** [1], a VAR-style model with a markedly different hybrid autoregressive attention design. As shown in the new HART results below (Tables 1), ToProVAR maintains comparable or slightly improved GenEval scores relative to the HART baseline, while significantly reducing inference latency. Moreover, compared with FastVAR, ToProVAR consistently achieves a better quality–efficiency trade-off on HART. These observations indicate that our tri-dimensional entropy-aware sparsity optimization is not tied to Infinity-specific architecture details and can transfer to other VAR backbones such as HART. We will further explore STAR-style architectures as part of future work.
>
> **Table 1: Comparison of ToProVAR and FastVAR on the GenEval benchmark using HART.**
>
> | Method        | Two Obj. | Position | Color  Attri. | Overall↑ | Latency (s)↓ | Speedup↑ |
> |---------------|----------|----------|---------------|----------|--------------|----------|
> | HART          | 0.62     | 0.13     | 0.18          | 0.51     | 0.95s        | 1.0×     |
> | +FastVAR      | 0.59     | 0.13     | 0.19          | 0.50     | 0.64s        | 1.5×     |
> | +ToProVAR     | 0.61     | 0.13     | 0.18          | 0.51     | 0.56s        | 1.7×     |
>
>
> [1] **HART: Efficient Visual Generation with Hybrid Autoregressive Transformer.**
>
> > **Q2:** The paper does not provide a detailed analysis of the computational overhead of SVD. Is there an analysis of the layer representation score? What are the patterns in the emergence of Global Layers and Detail Layers? Figure 10 shows an alternating pattern, is this behavior general?
>
> A2: Thank you for your detailed and insightful comments on the computational overhead of SVD and the behavior of Global vs. Detail Layers.
>
> 1. **Computational overhead of SVD.**
>    We agree that the SVD cost deserves clarification. In the revised manuscript **Appendix A.4.2**, we explicitly state that SVD is computed **once per layer at a single representative scale** (e.g., scale $s = 6$), rather than at all scales. The measured SVD time is **0.465 ms per layer**, and **13.96 ms** in total at that scale, which accounts for **less than 3%** of the end-to-end inference latency for Infinity-2B + ToProVAR (**610 ms**).
>
> **Table 2: Time cost of SVD.**
> | Operation                         | Time cost (ms) ↓ |
> | --------------------------------- | ---------------- |
> | SVD per layer at scale 6          | 0.4650           |
> | SVD (all layers) at scale $s = 6$ | 13.960           |

---

> ### Author Response · Authors · 2025-11-28
> **Response to Reviewer 2xuz(2/2)**
>
> 2. **Analysis of the layer representation score and layer patterns.**
>    As discussed in Eq. (4), we first compute the ratio
>    $\rho^{(l,s)} = \sigma^{(l,s)}_1 / \sigma^{(l,s)}_2$
>    between the largest and second largest singular values, and then map it to a layer representation score $\mathcal{R}^{(l,s)} \in (0, 1]$. Empirically, we observe a clear polarization: **Global Layers** exhibit a dominant principal component with $\rho^{(l,s)} \gg 1$ (thus $\mathcal{R}^{(l,s)} \to 0$), while **Detail Layers** have $\rho^{(l,s)} \approx 1$ (thus $\mathcal{R}^{(l,s)} \to 1$). This enables a robust binary classification into Global (0) and Detail (1) layers.
>
>    To answer your question about whether the alternating pattern in Figure 10 is general, we conducted a systematic analysis across scales $s = 4 \dots 13$ and multiple prompts. As summarized in Table 3, the **layer-type patterns are highly consistent across scales and prompts**: the first few layers and the very last layers are almost always classified as Global, while the middle layers are predominantly Detail. The cross-prompt similarity of the binary layer-type vectors is ≥ 0.90 at all scales, indicating that the observed “alternating” behavior is a **stable property of the VAR architecture**, rather than an artifact of a specific image.
>
> **Table 3: Layer-type classification and cross-prompt similarity at different scales (Infinity-2B, 32 layers).**
>
> | Scale | Layer type pattern (0: Global, 1: Detail)                                                                 | Cross-prompt similarity ↑ |
> |-------|-----------------------------------------------------------------------------------------------------------|---------------------------|
> | 4     | [0, 0, 0, 0, 1, 0, 1, 1, 1, 1, 1, 1, 1, 1, 1, 1, 1, 1, 1, 1, 0, 1, 1, 0, 0, 0, 1, 0, 1, 1, 0, 0]          | 0.87                      |
> | 5     | [0, 0, 0, 0, 1, 0, 1, 1, 1, 1, 1, 1, 1, 1, 1, 1, 1, 1, 1, 1, 0, 1, 1, 0, 0, 0, 1, 0, 1, 1, 0, 0]          | 0.86                      |
> | 6     | [0, 0, 0, 0, 1, 0, 1, 1, 1, 1, 1, 1, 1, 1, 1, 1, 1, 1, 1, 0, 0, 1, 1, 0, 1, 0, 1, 0, 1, 1, 0, 0]          | 0.91                      |
> | 7     | [0, 0, 0, 0, 1, 0, 1, 1, 1, 1, 1, 1, 1, 1, 1, 1, 1, 1, 1, 0, 0, 1, 1, 0, 1, 0, 1, 0, 1, 1, 0, 0]          | 0.90                      |
> | 8     | [0, 0, 0, 0, 1, 0, 1, 1, 1, 1, 1, 1, 1, 1, 1, 1, 1, 1, 1, 0, 0, 1, 1, 0, 1, 0, 1, 0, 1, 1, 0, 0]          | 0.91                      |
> | 9     | [0, 0, 0, 0, 1, 0, 1, 1, 1, 1, 1, 1, 1, 1, 1, 1, 1, 1, 1, 0, 0, 1, 1, 0, 1, 0, 1, 0, 1, 1, 0, 0]          | 0.95                      |
> | 10    | [0, 0, 0, 0, 1, 0, 1, 1, 1, 1, 1, 1, 1, 1, 1, 1, 1, 1, 1, 0, 0, 1, 1, 0, 1, 0, 1, 0, 1, 1, 0, 0]          | 0.94                      |
> | 11    | [0, 0, 0, 0, 1, 1, 1, 1, 1, 1, 1, 1, 1, 1, 1, 1, 1, 1, 1, 0, 0, 0, 1, 0, 1, 0, 1, 0, 1, 1, 0, 0]          | 0.95                      |
> | 12    | [0, 0, 0, 0, 1, 1, 1, 1, 1, 1, 1, 1, 1, 1, 1, 1, 1, 1, 1, 0, 0, 0, 1, 0, 1, 0, 1, 0, 1, 1, 0, 0]          | 0.97                      |
> | 13    | [0, 0, 0, 0, 1, 1, 1, 1, 1, 1, 1, 1, 1, 1, 1, 1, 1, 1, 1, 0, 0, 0, 1, 0, 1, 0, 1, 0, 1, 1, 0, 0]          | 0.96                      |
>
> We incorporated this analysis into the revised **Appendix A.4.2** to make the SVD cost and the generality of Global/Detail Layer patterns more explicit.
>
> > **Q3:** According to Table 4, Layer Representation Identification contributes most significantly to preserving generation quality (GenEval ↑ from 0.477 to 0.679), suggesting its critical role. Thus, more analysis such as qualitative cases would strengthen the paper’s claims.
>
> A3: Thank you for this insightful suggestion. In the revised manuscript, we have added qualitative visualizations in **Appendix A.6.4“Visualization of Ablation Study”** to directly illustrate the critical role of **Layer Representation Identification**. These new examples compare generations with and without this component, showing how it helps preserve global structure and fine-grained details.

---

### Official Review · Reviewer_TaTd · 2025-11-01

**Soundness:** 3
**Presentation:** 3
**Contribution:** 3
**Rating:** 6
**Confidence:** 2

**Summary:**

This paper introduces ToProVAR, which is a training free acceleration framework for VAR that uses attention-entropy to drive sparsity decisions jointly across tokens, layers, and scales. The method authors proposed can generalize entropy beyond local tokens to cross-layer and cross-scale scopes. ToProVAR has shown good performance, where on infinity-2b/8b, it reports 3.4x and 2.7-3.0x speedups respectively with little quality loss on GenEval/DPG/HPSv2/ImageReward and MJHQ30K.

**Strengths:**

$\bullet$ New sparsity signal: moves from frequency heuristics to entropy based semantic salience, and the results with illustrations are convincing.

$\bullet$ three dimension design is well structured: starting with low entropy ratio, layer classification via principal component ratio from SVD, then unified token retention probability.

$\bullet$ Nontrivial novelty in the engineering design: FAE integrates entropy into flash attention to avoid $N \times N$ instantiation and keep the linear-time behavior at the same time.

$\bullet$ Solid Empirical performance

**Weaknesses:**

$\bullet$ The paper takes low attention-entropy as salient, but there is no proof it correlates with task loss or semantics, nor normalization across heads and layers with different logit temperatures.

$\bullet$ Deriving the error bounds or optimality for the tri-stage greedy pruning: Scale -> layer -> tokens decisions are locally heuristic with no suboptimality gap, stability, or compounding-error control.

I understand this work focuses on empirical contributions, but it is often necessary to justify the empirical findings with explanations. I am not asking the authors to develop a full theoretical framework to address these weaknesses; I will be satisfied if the authors can answer these questions in text.

**Questions:**

$\bullet$ Can authors clarify the intended scope of attention-entropy as a salience proxy? How is it calibrated across heads/layers?

$\bullet$ Can authors discuss interactions across the scale -> layer -> token stages, provide the fallbacks, stop conditions, and outline a safe operating range where compounding errors are unlikely?

---

> ### Author Response · Authors · 2025-11-28
> **Response to Reviewer TaTd**
>
> We sincerely appreciate the valuable feedback and constructive suggestions. Thanks so much for taking time and effort to review our paper.
>
> > **Q1:** The paper takes low attention-entropy as salient, but there is no proof it correlates with task loss or semantics, nor normalization across heads and layers with different logit temperatures. Can authors clarify the intended scope of attention-entropy as a salience proxy? How is it calibrated across heads/layers?
>
> A1: Thank you for pointing this out. In our work, **attention entropy is used as a local, model-internal heuristic**, not as a theoretically proven surrogate for task loss. Given query (q_i) and keys (k_j), we compute attention weights $\alpha_{i,j} \propto \exp(q_i^\top k_j / \sqrt{d_k})$ and entropy $\mathcal{H}(q_i) = -\sum_j \alpha_{i,j}\log\alpha_{i,j}$; low entropy means attention mass is concentrated on a few positions, which we interpret as higher semantic selectivity. This assumption is **empirically** supported by token-pruning ablations: pruning high-entropy tokens degrades perceptual quality much less than pruning low-entropy ones, and by consistent qualitative patterns in our visualizations. For calibration, we (i) first **average attention over heads** and compute entropy on the head-averaged distribution, (ii) use **relative** entropy statistics per scale (the low-entropy ratio $\rho_s$ with respect to that scale’s mean) to decide the pruning start depth, (iii) use **SVD-based ratios** of the entropy map per layer/scale to derive a normalized layer score $R(l,s)$ that is invariant to global rescaling, and (iv) **normalize entropies within each (layer, scale)** when computing token-level gates. Thus all three stages rely on *relative* patterns of entropy within a fixed backbone, rather than raw cross-layer magnitudes.
>
>
> > **Q2:** Deriving the error bounds or optimality for the tri-stage greedy pruning: Scale -> layer -> tokens decisions are locally heuristic with no suboptimality gap, stability, or compounding-error control. Can authors discuss interactions across the scale -> layer -> token stages, provide the fallbacks, stop conditions, and outline a safe operating range where compounding errors are unlikely?
>
> A2: We agree that the tri-stage scheme is greedy and we do **not** claim formal optimality guarantees. Our design goal is a practically safe operating regime where errors do not compound, which we ensure structurally and empirically. Structurally, the three stages form a **nested refinement**: (i) the scale-level statistic $\rho_s$ only determines a start scale (D), so **no pruning ever happens on early coarse scales**; (ii) the layer-level score $R(l,s)$ separates Global vs. Detail layers, and **Global layers are pruned with a very small weight**; (iii) token-level pruning is applied only within selected Detail layers and is controlled by a bounded gate with $\alpha_{\min},\alpha_{\max}$, which caps the maximum sparsity and enforces a minimum keep ratio per region. As fallbacks/stop conditions, if $\rho_s$ never exceeds the threshold $\tau$ we skip scale-level pruning; if $R(l,s)$ indicates ambiguous or highly global behavior, the layer is treated conservatively (minimal pruning); and token-level gates never drop all tokens in any block. We further discuss the error bounds and optimality of the tri-stage greedy scheme in the **Appendix A.8**.

---

### Official Review · Reviewer_myfg · 2025-11-01

**Soundness:** 3
**Presentation:** 3
**Contribution:** 3
**Rating:** 4
**Confidence:** 2

**Summary:**

The authors show that attention entropy can be generalized across three dimensions (token, layer, and scale) to guide semantic-aware acceleration in visual autoregressive (VAR) models based on next-scale prediction. They first quantify semantic fineness across scales via the low-entropy ratio, determining the depth at which generation can safely begin pruning. Next, the authors classify layers into Global vs. Detail using the principal-component dominance (SVD-based ratio) of attention entropy, pruning only the Detail layers. Finally, the authors apply token-level salience pruning within prunable layers via a unified gating function that combines normalized entropy, layer scope, and scale depth. Empirically, the paper shows that compressing Global layers harms structure, whereas up to 90% compression on Detail layers preserves fidelity. Images with complex local detail show lower entropy and thus require deeper scales.

**Strengths:**

* Integrated semantic perspective. The tri‑dimensional entropy viewpoint connects token salience, layer scope, and multi‑scale semantics, which motivates where to prune.

* Layer taxonomy with a quantitative analysis. SVD‑based gives a reproducible criterion to distinguish Global/Detail layers before pruning.

* Scale‑aware depth selection. The low‑entropy ratio offers a principled way to adapt depth to content complexity rather than using a fixed scale budget.

* Empirical validation. The layer-specific compression curves (Fig. 3b) and ablations (Table 4) quantitatively support the semantic analysis, showing that pruning Detail layers up to 90% yields minimal quality degradation.

**Weaknesses:**

I'm not an actual expert in this area, but I have some concerns about the paper (including some appendix) based on my understanding.

* In terms of overhead and practicality, computing entropy per token, SVD per layer × scale, and tri‑dimensional gating online can be expensive. What is the actual (e.g., net) wall‑clock speedup vs. simpler frequency‑based methods once analysis overhead is included?

* Calibration burden: Depth threshold $\tau$ is selected by pre‑sampling (in Appendix). How robust is $\tau$ across prompts, resolutions, and models? Is there an online estimator to avoid pre‑runs?

* I think that the ablation studies should be done across dimensions. Could the authors isolate the incremental contributions of scale‑level, layer‑level, and token‑level pruning (three‑way ablation)? Current evidence is mostly qualitative.

* Generalization beyond Infinity‑series: Do the layer‑scope statistics and ρs behavior hold for other VAR families and codebooks? Please include failure cases where Global/Detail separation is ambiguous.

**Questions:**

Please refer to the weaknesses.

---

> ### Author Response · Authors · 2025-11-28
> **Response to Reviewer myfg(1/4)**
>
> Thank you very much for your positive feedback and constructive suggestions. Your professional advice has been invaluable in further improving our work!
>
> > **Q1:** In terms of overhead and practicality, computing entropy per token, SVD per layer $\times$ scale, and tri‑dimensional gating online can be expensive. What is the actual (e.g., net) wall‑clock speedup vs. simpler frequency‑based methods once analysis overhead is included?
>
> A1: Thank you very much for your thoughtful question on the computational overhead and net wall-clock speedup of our method. We address this from both the *micro-level operator cost* and the *end-to-end latency* perspective.
>
> 1. **Cost of entropy and SVD computations.**
>    In the revised manuscript, we add a systematic cost analysis in **Appendix A.4 (“Computational Cost Analysis”)**, where we separately measure the time of frequency-based scoring, naïve attention entropy, our integrated Flash Attention Entropy (FAE), and SVD.
>
>    - **Flash Attention Entropy (FAE):**
>      As reported in **Appendix A.4.1**, Naive attention entropy requires full materialization of the attention matrix and is therefore an order of magnitude slower than FlashAttention. In contrast, FAE computes the entropy on-the-fly inside the FlashAttention kernel, without explicitly constructing the attention matrix. The only substantial extra work comes from the row-wise $x \log x$ reduction in the innermost loop.
>     As shown in Table 1, its runtime over all scales on Infinity-8B is **12.96 ms**, compared to **11.27 ms** for plain FlashAttention and **125.73 ms** for a naïve attention-entropy implementation. Thus, FAE adds only a small absolute overhead (1.69 ms) per layer over plain FlashAttention,, while reducing the entropy computation cost by roughly **90%** compared to naïve Attention Entropy.
>
>
>    - **SVD:**
>      In the revised manuscript we clarify that SVD is computed **once per layer at a single representative scale** (e.g., scale $s = 6$), rather than at all scales. **Appendix A.4.2** reports a per-layer SVD time of **0.465 ms**, and a total SVD time of **13.96 ms** at that scale, which accounts for **less than 3%** of the overall inference latency for Infinity-2B + ToProVAR (**$\approx$610 ms**), as shown in Table 2..
>    - **Tri-dimensional gating:**
>      The tri-dimensional gating consists only of lightweight thresholding and simple arithmetic integrated into the decoding loop; its runtime overhead is negligible compared to attention and MLP layers.
>
>    **Table 1: Time cost of frequency / entropy-related operations (per layer).**
>
>    | **Operation/Time cost (ms)**                          | **All Scales** | **Scale 10** |
>    | ----------------------------- | ------------------------------------------- | -------- |
>    | Frequency                     | 1.30                                        | 0.16     |
>    | Attention Entropy (naïve)     | 125.73                                      | 12.06    |
>    | FlashAttention                | 11.27                                       | 1.11     |
>    | Flash Attention Entropy (FAE) | 12.97                                       | 1.28     |
>
>    **Table 2: Time cost of SVD.**
>
>    | Operation                         | Time cost (ms) ↓ |
>    | --------------------------------- | ---------------- |
>    | SVD per layer at scale 6          | 0.4650           |
>    | SVD (all layers) at scale $s = 6$ | 13.960           |
> 2. **Net wall-clock speedup vs. frequency-based methods.**
>    Regarding the net wall-clock effect “once analysis overhead is included,” we report **end-to-end latency** in the main paper (5.2 Main Result). On Infinity-2B, our method achieves a **3.4× net speedup over the base model** and about **1.3–1.4× higher net speedup than the frequency-based FastVAR baseline**. These results indicate that, even after accounting for entropy, SVD, and gating overhead, ToProVAR still provides substantial **net** acceleration over both the plain Infinity model and the simpler frequency-based FastVAR, while preserving generation quality.

---

> ### Author Response · Authors · 2025-11-28
> **Response to Reviewer myfg(2/4)**
>
> > **Q2:** Calibration burden: Depth threshold $\tau$ is selected by pre‑sampling (in Appendix). How robust is $\tau$  across prompts, resolutions, and models? Is there an online estimator to avoid pre‑runs?
>
> A2: Thank you for raising this important question about the robustness and calibration of the depth threshold $\tau$. In the revised manuscript we have added a dedicated robustness study in **Appendix A.5 (“Calibration and Robustness of the Scale-Depth Threshold $\tau$”)**.
>
> 1. **Robustness of $\rho_s$ and the choice of a global $\tau$.**
>    Building on the scale-level analysis in Eq. (3) and Fig. 4(a), we examine the scale-wise low-entropy ratio
>    $\rho_s = \frac{\lvert { i \mid H_i^s < \bar{H}^s } \rvert}{N_s}$,
>    and find that for a given backbone, the **mean entropy $\bar{H}^s$ and $\rho_s$ curves are highly consistent across different datasets and calibration subsets**. Concretely, for both Infinity-2B and Infinity-8B, we:
>
>    * sample a small calibration subset of **50 prompts** from **HPSv2.1** and **DPG-Bench**,
>    * compute $\bar{H}^s$ and $\rho_s$ across scales, and
>    * compare these curves to those obtained using the full prompt sets.
>
>    As shown in **Appendix A.5**, the curves for HPSv2, DPG-Bench, and the 50-prompt subsets nearly overlap, indicating that attention-entropy statistics are largely **insensitive to the dataset and calibration size**.
>
>    To further validate robustness, we also examine the distribution of $\rho_s$ for images whose SSIM to the full-scale baseline exceeds 0.8. These high-quality generations cluster around a narrow band centered at **$\rho_s \approx 0.4$**, which we use as a quality-aware indicator for semantic stability. This motivates setting a **single global threshold $\tau = 0.4$ per backbone**, and in practice we use **$\tau = 0.4$ for all Infinity-2B/8B experiments**.
>
>    In summary, $\tau$ is calibrated **once per backbone** on a small calibration subset, and then **reused across all prompts and resolutions**, without any dataset-specific hyperparameter search.
>
> 2. **On the possibility of avoiding pre-runs via an online estimator.**
>    We appreciate the reviewer’s suggestion to use an online estimator for $\tau$. However, in our preliminary explorations, such online criteria tended to introduce **additional hyperparameters and non-negligible runtime overhead** (e.g., per-sample convergence checks during generation). Therefore, in the current version we adopt a **single offline pre-sampling step** (lightweight and executed only once per model) as a practical compromise, and leave the design of a more sophisticated online estimator for $\tau$ as an interesting direction for future work (also mentioned in the Limitations/Future Work section, Appendix A.9).

---

> > ### Author Response · Authors · 2025-11-28
> > **Response to Reviewer myfg(3/4)**
> >
> > > **Q3:** I think that the ablation studies should be done across dimensions. Could the authors isolate the incremental contributions of scale‑level, layer‑level, and token‑level pruning (three‑way ablation)? Current evidence is mostly qualitative.
> >
> > A3: Thank you for pointing out the need for a more fine-grained, cross-dimensional ablation. ToProVAR is designed as a **progressive tri-dimensional optimization framework**: scale-, layer-, and token-level modules are tightly coupled and not fully independent. Therefore, our ablation focuses on *incrementally* adding each dimension to quantify both its individual contribution and its interaction with the others.
> >
> > In the main paper (Sec. 5.3), we already provide a three-stage ablation on Infinity-2B. To further isolate the contributions across dimensions, we introduce two additional controlled variants in the **extended ablation of Appendix A.3.5**. We summarize the results again below:
> >
> > 1. **Fix Scale Exit.(No Scale)** A coarse baseline that uses a fixed exit scale without Scale Depth Localization. This variant achieves the highest nominal speed (0.41 s, ≈5.1×) but the worst GenEval (0.403), underscoring the importance of *adaptive* scale selection.
> > 2. **Scale Depth Loc. + Fine-grained Token Prun. (Scale + Token).** Even with adaptive scales and token pruning, GenEval remains noticeably lower (0.603 vs. 0.690), highlighting that **unconstrained token pruning on Global layers can harm global semantics**.
> > 3. **Scale Depth Loc. + Layer Repr. Ident. (Scale + Layer).** Adding layer-level representation identification on top of scale-level pruning substantially recovers global structure and semantics (GenEval 0.679) while maintaining strong acceleration (3.7×).
> > 4. **Full ToProVAR (Scale + Layer + Token).** The complete tri-stage framework adds token-level sparsity **restricted to Detail layers**, restoring the GenEval score to 0.690—on par with the baseline—while keeping 3.4× speedup.
> >
> > Overall, these ablations demonstrate the **progressive improvements and complementary roles** of scale-, layer-, and token-level optimization in balancing acceleration and quality, and justify the full tri-dimensional design.
> >
> > **Table 3: Ablation study of the tri-dimensional progressive framework on Infinity-2B.**
> >
> > | Method                                                | Latency (s)↓ | Speed↑ | GenEval↑ |
> > |-------------------------------------------------------|--------------|--------|----------|
> > | Infinity-2B                                           | 2.10         | 1.0×   | 0.690    |
> > | Fix Scale Exit                                        | 0.41         | 5.1×   | 0.403    |
> > | Scale Depth Loc.                                      | 0.47         | 4.5×   | 0.477    |
> > | Scale Depth Loc. + Fine-grained Token Prun.           | 0.78         | 2.7×   | 0.603    |
> > | Scale Depth Loc. + Layer Repr. Ident.                 | 0.57         | 3.7×   | 0.679    |
> > | ToProVAR                                              | 0.61         | 3.4×   | 0.690    |

---

> ### Author Response · Authors · 2025-11-28
> **Response to Reviewer myfg(4/4)**
>
> > **Q4:** Generalization beyond Infinity‑series: Do the layer‑scope statistics and $\rho_s$ behavior hold for other VAR families and codebooks? Please include failure cases where Global/Detail separation is ambiguous.
>
> A4: Thank you for raising this important question about generalization beyond the Infinity series. First, we observe that the proposed layer-scope statistics and the scale-wise low-entropy ratio ρₛ extend naturally to other VAR families. In the revised manuscript, we add **layer-level visualizations for HART** in **Appendix A.6.2 “Layer-Level Semantic Representation Visualizations”**, where HART exhibits a very similar pattern: early and final layers act as Global Layers, while middle layers behave as Detail Layers with more localized semantics. To further assess generality at the model level, we also evaluate ToProVAR on **HART** [1]. As shown in Table 4, ToProVAR maintains comparable GenEval scores to the HART baseline while significantly reducing latency, and consistently offers a better quality–efficiency trade-off than FastVAR on HART.
>
> Regarding **failure cases where Global/Detail separation is ambiguous**, we thank the reviewer for raising this point. After revisiting our visualizations, we indeed observe a small number of ambiguous cases where the separation is not perfectly binary.
> In **Appendix A.6.2**, we further illustrates the rare ambiguous cases. In these layers, the attention maps exhibit both weak grid-like global structure and pronounced object-centric activations, so that the layer plays a mixed role of refining global layout and local details. Such hybrid behavior typically arises at transition layers where global composition is being finalized while fine details start to emerge, and is further amplified by averaging over heads, since different heads may specialize in global versus local semantics. Consequently, these layers lie close to the decision boundary of our principal-component–based classifier and can be labeled as Global or Detail depending on small variations across prompts or scales. However, they account for only a very small subset of layers we inspected, and their impact on final accuracy is negligible.
>
> **Table 4: Comparison of ToProVAR and FastVAR on the GenEval benchmark using HART.**
>
> | Method    | Two Obj. | Position | Color | Attri. | Overall↑ | Latency (s)↓ | Speedup↑ |
> | --------- | -------- | -------- | ----- | ------ | -------- | ------------ | -------- |
> | HART      | 0.62     | 0.13     | 0.18  | 0.51   | 0.51     | 0.95         | 1.0×     |
> | +FastVAR  | 0.59     | 0.13     | 0.19  | 0.50   | 0.50     | 0.64         | 1.5×     |
> | +ToProVAR | 0.61     | 0.13     | 0.18  | 0.51   | 0.51     | 0.56         | 1.7×     |
>
> [1] **HART: Efficient Visual Generation with Hybrid Autoregressive Transformer.**

---

### Author Response · Authors · 2025-12-04
**Summary of Reviews and Author Responses**

Dear Area Chair,

We appreciate your time and effort in handling our submission. As the rebuttal phase is coming to an end, we would like to briefly summarize the context to aid you in making the final decision.

## **Summary of the Reviews**
All four reviewers evaluated our work positively on soundness, presentation, and contribution (“good”). Two reviewers gave overall ratings of 8 (poster), one gave 6 (marginally above acceptance), and the remaining reviewer provided a positive assessment. They agree that ToProVAR is a well-motivated, training-free, tri-dimensional entropy-based acceleration framework for VAR models, with solid empirical gains (≈3.4–3.5× speedup with minor quality loss) and a non-trivial engineering contribution (Flash Attention Entropy).

The main concerns can be grouped as follows:

* **Overhead and calibration (Reviewers myfg, 2xuz, FVpx).**
  Net wall-clock speedup when including entropy/SVD/gating overhead and hyperparameter calibration; computational cost of FlashAttn-Entropy and SVD; robustness and tuning burden of the scale-depth threshold τ; cost of one-time calibration.

* **Design analysis and theory (Reviewers TaTd, myfg).**
  Need for clearer three-way ablations across scale / layer / token; interactions and stability of the tri-stage greedy scheme (scale → layer → token), including fallbacks and safe operating regimes; clarification of attention entropy as a salience proxy and its calibration across heads/layers.

* **Generalization and diagnostics (Reviewers myfg, 2xuz, FVpx).**
  Evaluation on additional backbones beyond Infinity (e.g., HART, STAR-style models); deeper analysis of SVD-based layer representation scores, Global vs. Detail layer patterns, and ambiguous or hybrid cases.

* **Presentation issues (Reviewers FVpx, 2xuz).**
  Additional qualitative illustrations, clearer figure legends, and minor table formatting corrections.

## **Our Responses and Revisions in the Updated Manuscript**
Our rebuttal and revised manuscript systematically address these concerns through new analyses, experiments, and clarifications:

* We provide a detailed **computational cost analysis** (Appendix A.4), showing that Flash Attention Entropy incurs only 0.17 ms overhead over FlashAttention at scale 10 while being ~90% faster than naïve entropy, SVD contributes <3% of end-to-end latency, and ToProVAR still achieves ≈3.4× net speedup over the base model and ≈1.3–1.4× over FastVAR.

* We conduct a **robustness study of the scale-depth threshold τ** (Appendix A.5), demonstrating that entropy statistics and the low-entropy ratio ρₛ are stable across datasets and small calibration subsets. A single τ per backbone, calibrated once on 50 prompts with modest cost (~0.3 GPU-hours), is reused across prompts, datasets, and resolutions.

* We extend **tri-dimensional ablations and interaction analysis** (Appendix A.3.5, A.8), quantifying the incremental effects of scale-, layer-, and token-level pruning and explaining the nested, conservative design with explicit fallbacks and bounded token gates to avoid compounding errors.

* We clarify the **intended scope and calibration of attention entropy** as an empirical salience heuristic, detailing head-averaging and relative normalization at scale, layer, and token levels rather than relying on raw magnitudes.

* We add **generalization experiments on HART** (showing that ToProVAR maintains baseline-level quality with improved speed and consistently outperforms FastVAR in quality–efficiency trade-offs) and broaden the **layer representation analysis** (Appendix A.4.2, A.6.2), including consistent Global/Detail patterns across scales and prompts, as well as visualizations of rare ambiguous layers.

* We improve **presentation quality** by adding legends to the relevant figures, correcting boldface in Table 1, and including new qualitative visualizations of ablations (Appendix A.6.4) to illustrate the critical role of Layer Representation Identification.

We sincerely appreciate the constructive feedback from all reviewers and believe that our responses and revisions substantially strengthen the paper and comprehensively address the core concerns.

Best wishes,

Authors of Submission 1419

---

### Meta-Review · Area_Chair_NizE · 2026-01-06

**Summary:**

The paper introduces ToProVAR, which is a training free acceleration framework for VAR, using uses attention-entropy to drive sparsity decisions jointly across tokens, layers, and scales.

The reviewers concerns on the paper lies in the following
1. The computational overhead  [Reviewer myfg, Reviewer FVpx]
2. Detailed ablation studies [Reviewer myfg], including additional backbones beyond Infinity [Reviewer 2xuz, FVpx], analysis on the layer representation indentification [Reviewer 2xuz]
3. Justification of the empirical findings. [Reviewer TaTd]


Most of the reviewers express postive opinions on the acceptance of the paper. The paper does have technical novelties, although missing some detailed analysis. The paper does have merits. As such, I am considering to accept the paper

**Reviewer Concerns:**

The computational overhead has been justified.
Detailed ablation studies can be addressed by the rebuttal.
The jsutfication of the empoircal finding is outstanding. However, it does not affection the novelties and contributions of the paper.

**Reviewer Scores:**

NA

---

### Decision · Program_Chairs · 2026-01-26

Accept (Poster)